# UniKP: a unified framework for the prediction of enzyme kinetic parameters

Han Yu [1,2,3,4,9], Huaxiang Deng[1,3,4,9], Jiahui He[1,3,4], Jay D. Keasling [4,5,6,7,8] & Xiaozhou Luo [1,2,3,4] ✉

Prediction of enzyme kinetic parameters is essential for designing and optimizing enzymes for various biotechnological and industrial applications, but the limited performance of current prediction tools on diverse tasks hinders their practical applications. Here, we introduce UniKP, a unified framework based on pretrained language models for the prediction of enzyme kinetic parameters, including enzyme turnover number ($k_{cat}$), Michaelis constant ($K_m$), and catalytic efficiency ($k_{cat} / K_m$), from protein sequences and substrate structures. A two-layer framework derived from UniKP (EF-UniKP) has also been proposed to allow robust $k_{cat}$ prediction in considering environmental factors, including pH and temperature. In addition, four representative re-weighting methods are systematically explored to successfully reduce the prediction error in high-value prediction tasks. We have demonstrated the application of UniKP and EF-UniKP in several enzyme discovery and directed evolution tasks, leading to the identification of new enzymes and enzyme mutants with higher activity. UniKP is a valuable tool for deciphering the mechanisms of enzyme kinetics and enables novel insights into enzyme engineering and their industrial applications.

The study of enzyme catalysis efficiency to a specific substrate is a fundamental biological problem that has a profound impact on enzyme evolution, metabolic engineering, and synthetic biology[1–3]. The in vitro measured values of $k_{cat}$ and $K_m$, the maximal turnover rate and Michaelis constant, are the indicators of the efficiency of an enzyme in catalyzing a specific reaction and can be used to compare the relative catalytic activity of different enzymes[4]. Currently, the measurement of the enzyme kinetic parameters relies primarily on experimental measurements, which are time-consuming, costly, and labor-intensive, resulting in a small database of experimentally measured kinetic parameters values[5]. For instance, the sequence database UniProt contains over 230 million enzyme sequences, while enzyme databases BRENDA and SABIO-RK contain tens of thousands of experimentally measured $k_{cat}$ values[6–8]. The integration of Uniprot identifiers in these enzyme databases has facilitated the connection between measured parameters and protein sequences. However, these connections are still far smaller in scale compared to the number of enzyme sequences, limiting the advancement of downstream applications such as directed evolution and metabolic engineering.

Researchers have attempted to utilize computational methods to accelerate the process of enzyme kinetic parameters prediction, but current approaches have exclusively concentrated on addressing one

[1]Shenzhen Key Laboratory for the Intelligent Microbial Manufacturing of Medicines, Shenzhen Institute of Advanced Technology, Chinese Academy of Sciences, Shenzhen 518055, China. [2]University of Chinese Academy of Sciences, Beijing 100049, China. [3]CAS Key Laboratory of Quantitative Engineering Biology, Shenzhen Institute of Synthetic Biology, Shenzhen Institute of Advanced Technology, Chinese Academy of Sciences, Shenzhen 518055, China. [4]Center for Synthetic Biochemistry, Shenzhen Institute of Synthetic Biology, Shenzhen Institute of Advanced Technology, Chinese Academy of Sciences, Shenzhen 518055, China. [5]Joint BioEnergy Institute, Emeryville, CA 94608, USA. [6]Biological Systems and Engineering Division, Lawrence Berkeley National Laboratory, Berkeley, CA 94720, USA. [7]Department of Chemical and Biomolecular Engineering & Department of Bioengineering, University of California, Berkeley, CA 94720, USA. [8]Novo Nordisk Foundation Center for Biosustainability, Technical University of Denmark, 2800, Kgs Lyngby, Denmark. [9]These authors contributed equally: Han Yu, Huaxiang Deng. ✉e-mail: xz.luo@siat.ac.cn

of these issues, overlooking the similarity of both tasks in reflecting the relationship of protein sequences towards substrate structures. A statistical approach has been proposed to infer $K_m$ values across species based on the known parameters of related enzymes[9]. An organism-independent model has been built which successfully predicts $K_m$ values for natural enzyme–substrate combinations using machine learning and deep learning[10]. Machine learning has also been demonstrated to be able to predict catalytic turnover numbers in *Escherichia coli* based on enzyme biochemistry, protein structure, and network context[11]. In addition, a deep learning-based method for $k_{cat}$ prediction from substrate structures and protein sequences has been developed, enabling high-throughput prediction[12]. Despite various prediction tools for $k_{cat}$ and $K_m$, they often fail to accurately capture the relationship between these two parameters. Consequently, the calculated $k_{cat} / K_m$ values from these models often deviate significantly from the experimental measurements. This discrepancy highlights the importance of a demonstration of a unified method for calculating or predicting $k_{cat} / K_m$, which is a crucial parameter reflecting catalytic efficiency. As a result, this has hindered the practical application of these methods in biotechnological and industrial contexts. Additionally, current models have failed to account for environmental factors such as pH and temperature, which can significantly impact enzyme kinetics[13]. Furthermore, the present models struggle to address high-value prediction problems due to imbalanced datasets, despite their importance in various biological research applications[14,15]. All these limitations have confined the current use of these tools to data analysis and model development, with no substantial impact on practical challenges such as enzyme discovery and directed evolution, which are of significant relevance to associated fields.

To overcome these limitations, a unified enzyme kinetic parameters prediction model with high accuracy needs to be developed. Recent breakthroughs in deep learning, particularly in the area of unsupervised learning from natural language processing, have led to novel data representation approaches that have been applied to biological problems with great success[16–20]. The advancement of pretrained language models for proteins and small molecules, represented using SMILES notation[18–20], has illuminated the path towards a more effective model for predicting enzyme kinetic parameters. The absence of datasets documenting the impact of environmental factors on kinetic parameters constitutes a major impediment to the prediction of parameter values under these conditions. Constructing a dataset that encompasses this information, and employing a two-layer ensemble model that integrates information from multiple models[12,21], is crucial to enhance the accuracy of these predictions. The distribution of experimentally measured kinetic parameters is imbalanced[12], characterized by a scarcity of high-value kinetic parameter samples, much like imbalanced and long-tailed datasets commonly encountered in the field of visual recognition, where a few categories are heavily populated, while most categories only contain a limited number of samples[22]. In visual recognition problems, researchers have utilized various techniques to improve the accuracy of the model in predicting high values[23,24]. One effective and straightforward approach is the re-weighting method, which enhances the significance of underrepresented categories by increasing their weight in the model[25].

Here, we present a pretrained language model-based enzyme kinetic parameters prediction framework (UniKP), which improves the accuracy of predictions for three enzyme kinetic parameters, $k_{cat}$, $K_m$, and $k_{cat} / K_m$, from a given enzyme sequence and substrate structure. We conducted a comprehensive comparison of 16 diverse machine learning models and 2 deep learning models, and demonstrated its remarkable improvement over previous prediction methods. Additionally, we have proposed a two-layer framework to consider environmental factors and validated its effectiveness on two representative datasets, including pH and temperature. We also applied typical re-weighting methods to the $k_{cat}$ dataset and successfully reduced the error of high value prediction. Lastly, we employed UniKP to assist the mining and directed evolution of tyrosine ammonia lyase (TAL), leading to the discovery of one TAL homolog from a database exhibiting significant enhanced $k_{cat}$, and the identification of two TAL mutants with the highest $k_{cat} / K_m$ values reported to date. We also validated that EF-UniKP consistently identifies highly active TAL enzymes with remarkable precision, when accounting for environmental factors.

## Results

### Overview of UniKP

The UniKP framework is comprised of two key components: a representation module and a machine learning module (Fig. 1). The representation module is used to encode information about enzymes and substrates using pretrained language models. Specifically, the amino acids in the enzyme sequence are transformed into a 1024-dimensional vector using the ProtT5-XL-UniRef50 model (Fig. 1a). Mean pooling is applied to obtain per-protein representation, which has been found to be the most effective method for per-protein tasks[18]. On the other hand, the substrate structure is converted to a simplified molecular-input line-entry system (SMILES) format and processed by the pretrained SMILES transformer, resulting in a 256-dimensional vector for each symbol. The mean and max pooling of the last layer and the first outputs of the last and penultimate layers are then concatenated to generate a per-molecular representation vector of 1024 dimensions[20] (Fig. 1b). The concatenated representation vector of both the protein and substrate are then fed into the following machine learning module. Here, the projections of t-distributed Stochastic Neighbour Embedding (t-SNE) with different perplexity and iterations demonstrated that a solely concatenated representation vector cannot discriminate well between high and low $k_{cat}$ values[26], further emphasizing the need for the machine learning model (Supplementary Fig. 1).

For the machine learning module, in order to explore the performance of different models, we conducted a comprehensive comparison of 16 diverse machine learning models, including basic linear regression to complex ensemble models, as well as 2 representative deep learning models, the Convolutional Neural Network (CNN) and the Recurrent Neural Network (RNN) (Fig. 2). Overall, the results showed that simpler models like linear regression displayed relatively poor prediction performance (Linear Regression $R^2 = 0.38$). In contrast, ensemble models demonstrated better performance. Notably, random forests and extra trees significantly outperformed other models, with extra trees exhibiting the highest performance (Extra Trees $R^2 = 0.65$). However, both categories of deep learning models, due to their demanding requirements for intricate network design and fine-tuning, did not perform as effectively in comparison (CNN $R^2 = 0.10$, RNN $R^2 = 0.19$). The results confirmed the significant advantage of ensemble models, with the extra trees model standing out as the best model. It's worth noting that the datasets are relatively small (~10 k) and the features are high-dimensional (2048d), making ensemble models more suitable. Simpler linear models exhibit lower fitting capability, while more complex neural networks require a large amount of labelled data, potentially making them unsuitable for this problem. Therefore, the concatenated representation vectors of both the protein and substrate are subsequently input into the interpretable extra trees model for the prediction of three distinct enzyme kinetic parameters (Fig. 1c).

Furthermore, to address two specific subproblems, we have fine-tuned the model for a better enzyme kinetic parameter prediction. One focuses on improving prediction performance by considering environmental factors (Fig. 1d). The other emphasizes optimization within the high-value range with higher error (Fig. 1e).

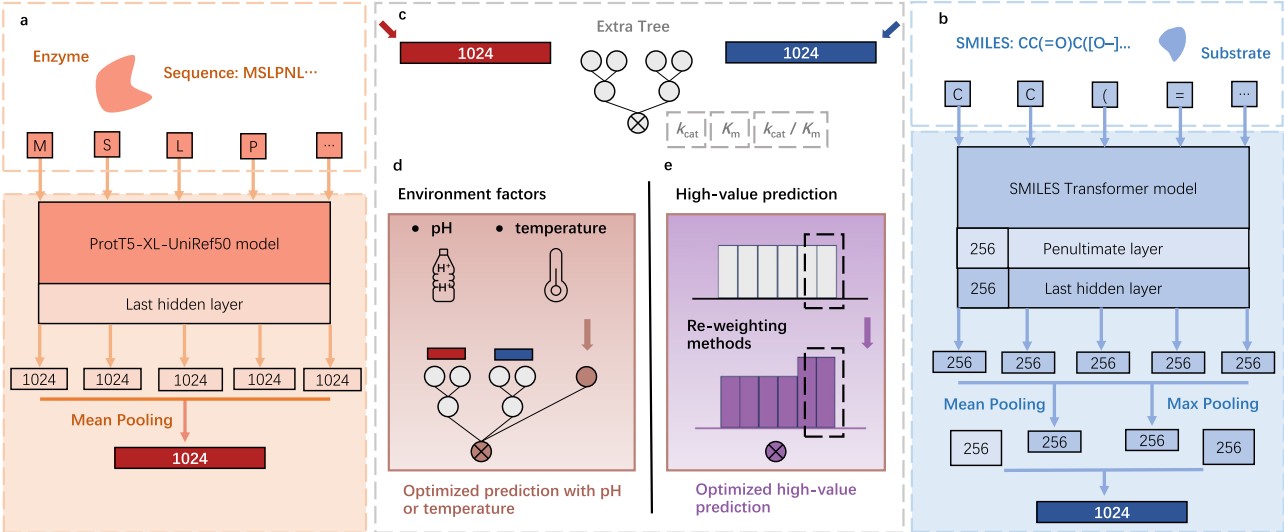

**Fig. 1 | The overview of UniKP. a** Enzyme sequence representation module: Information about enzymes was encoded using a pretrained language model, ProtT5-XL-UniRef50. Each amino acid was converted into a 1024-dimensional vector on the last hidden layer, and the resulting vectors were summed and averaged by mean pooling, generating a 1024-dimensional vector to represent the enzyme. **b** Substrate structure representation module: Information about substrates was encoded using a pretrained language model, SMILES Transformer model. The substrate structure was converted into a simplified molecular-input line-entry system (SMILES) representation and input into a pretrained SMILES transformer to generate a 1024-dimensional vector. This vector was generated by concatenating the mean and max pooling of the last layer, along with the first outputs of the last and penultimate layers. **c** Machine learning module: An explainable Extra Trees model took the concatenated representation vector of both the enzyme and substrate as input and generated a predicted $k_{cat}$, $K_m$ or $k_{cat}$ / $K_m$ value. **d** EF-UniKP: A framework that considers environmental factors to generate an optimized prediction. It is validated on two representative datasets: pH and temperature datasets. **e** Various re-weighting methods were used to adjust the sample weight distribution to generate an optimized prediction for high-value prediction task.

## High accuracy of UniKP in enzyme $k_{cat}$ prediction

We first validated our proposed framework, UniKP, on a $k_{cat}$ prediction task, using the DLKcat dataset, which comprises 16,838 samples. The results were compared with those of DLKcat using the same evaluation metrics in the original publication[12]. Without any additional optimization of parameters, the average coefficient of determination ($R^2$) value on the test set from five rounds of random splitting was 0.68, a 20% improvement over DLKcat (Fig. 3a). Additionally, the highest value of DLKcat in these five rounds was 16% lower than the lowest value of UniKP, further demonstrating the robustness of UniKP. The root mean square error (RMSE) between predicted and experimentally measured $k_{cat}$ values was also lower in UniKP compared to DLKcat, both in the training and test sets (Fig. 3b). We found a strong correlation between the predicted and experimentally measured $k_{cat}$ values in the test set (Fig. 3c; Pearson correlation coefficient (PCC) = 0.85) and the entire dataset (Supplementary Fig. 2a; PCC = 0.99), which were higher than those of DLKcat by 14% and 11%, respectively. Additionally, UniKP showed better prediction performance on a more stringent test set where either the enzyme or substrate was not present in the training set (Supplementary Fig. 2b; PCC = 0.83 vs. 0.70). UniKP also demonstrated superior performance in various $k_{cat}$ numerical intervals (Fig. 3d). In conclusion, UniKP outperforms DLKcat in enzyme $k_{cat}$ prediction across multiple evaluation metrics.

Furthermore, to demonstrate UniKP's ability to distinguish enzymes from different metabolic contexts, we separated the enzymes and their corresponding substrates into two categories, primary central and energy metabolism, and intermediary and secondary metabolism. Theoretically, the former should exhibit higher values[27,28]. Our results found that the primary central and energy metabolism category was significantly higher than the latter, consistent with expectations (Fig. 3e; $p = 9.33 \times 10^{-8}$). To gain insights into the model's learning process, we used SHapley Additive exPlanations (SHAP) to analyze feature importance[29]. Higher values indicate more significant features. We calculated the SHAP value of each enzyme and substrate feature in the test set based on the trained UniKP. Our findings showed that 15

out of the top 20 features belonged to the enzyme category, while the rest belonged to the substrate (Fig. 3f). This confirms that the embedded enzyme features are more critical than those of the substrate. Furthermore, among the top 20 features, 12 were positively correlated with predicted $k_{cat}$ values, and 8 were negatively correlated. These results indicate that UniKP has a distinct preference for enzyme features, underscoring the decisive role of enzyme information.

In order to evaluate the potential data leakage, we used a reported method to compare the performance of UniKP with the geometric mean of experimental data which represent the potential data leakage[30]. The result demonstrated a clear advantage (higher correlation and lower RMSE) of UniKP over the geometric means, indicating the absence of data leakage in the training process (Supplementary Fig. 3).

## UniKP markedly discriminates between $k_{cat}$ values of enzymes and their mutants

The ability to screen mutated enzymes is crucial in the process of enzyme evolution and its downstream applications. To further validate the discriminative power of UniKP, the dataset was segregated into wild-type and mutated enzymes based on DLKcat annotation[12]. The prediction results of UniKP were remarkable for both wild-type (Fig. 4a for the test set, PCC = 0.78; Supplementary Fig. 4a for the whole dataset, PCC = 0.98) and mutant enzymes (Fig. 4b for the test set, PCC = 0.91; Supplementary Fig. 4b for the whole dataset, PCC = 0.99). On the test set for both wild-type and mutant enzymes, the PCC of UniKP was found to be 13% higher than that of DLKcat (Fig. 4c). Moreover, we obtained $R^2$ values of 0.60 for wild-type enzymes and 0.81 for mutant enzymes, along with RMSE values of 0.90 for wild-type enzymes and 0.67 for mutant enzymes. In order to further validate the ability of the proposed UniKP framework, we conducted additional experiments. We randomly selected 5 groups of enzymes for each category of Enzyme Commission (EC) number, with each group consisting of two enzymes and their corresponding $k_{cat}$ values for one specific substrate[6]. Our framework was found to outperform DLKcat in

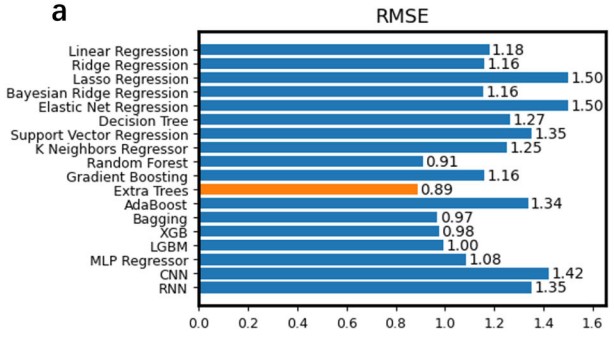

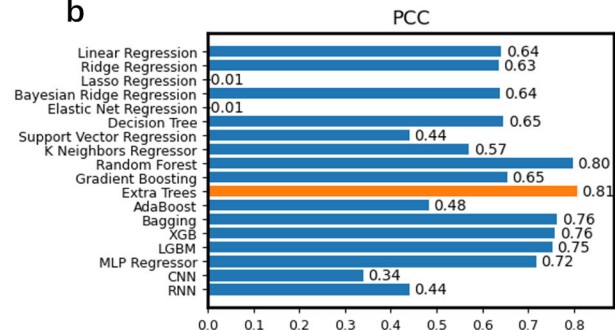

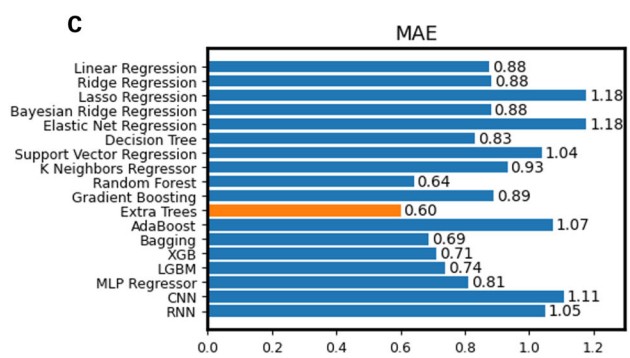

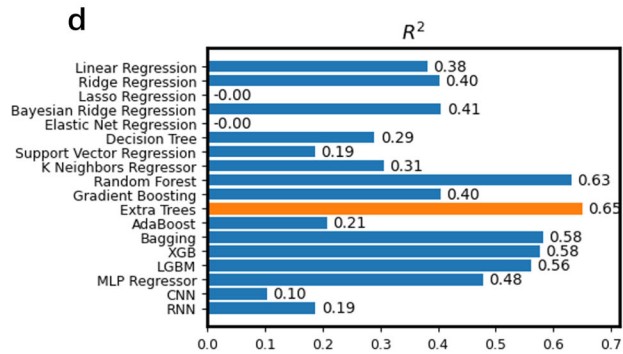

**Fig. 2 | Performance comparison of different models.** Comparison of Root Mean Square Error (RMSE) (**a**), Pearson Correlation Coefficient (PCC) (**b**), Mean Absolute Error (MAE) (**c**), and $R^2$ (Coefficient of Determination) (**d**) values between experimentally measured $k_{cat}$ values and predicted $k_{cat}$ values of 16 diverse machine learning models and 2 deep learning models. The $k_{cat}$ values of all samples were predicted independently using 5-fold cross-validation. Each bar in the graph represents the models' performance with respect to this metric. The "Extra Trees" model is highlighted in yellow, while other models are depicted in blue. The corresponding numerical values for each bar are provided on the right side. Source data are provided as a Source Data file.

differentiating the $k_{cat}$ values of different enzymes for a given substrate (Supplementary Fig. 4c; 80% vs 53%).

**A two-layer framework considering environmental factors**

In order to simulate a more realistic biological experiment and predict more accurate $k_{cat}$ values considering environmental factors, we proposed a two-layer framework called EF-UniKP (Fig. 5a). This strategy is based on a two-layer framework that considers the influence of environmental factors such as pH and temperature. The base layer of the framework consists of two individual models, namely UniKP and Revised UniKP. The UniKP was trained using a dataset without environmental information to predict a rough $k_{cat}$ value, while the Revised UniKP was trained using a smaller dataset that considered an arbitrary environmental factor, such as pH or temperature. The input of the Revised UniKP is a concatenated representation vector of the protein and substrate, in combination with the pH or temperature value. The meta layer of the framework is a linear regression model that takes the predicted $k_{cat}$ values from both the UniKP and Revised UniKP as inputs. By considering information from both datasets with and without environmental information, the final predicted $k_{cat}$ value is more accurate.

To prospectively evaluate our proposed strategy, we assessed the influence of pH and temperature on enzyme-substrate reactions by creating two datasets. These contained enzyme sequences, substrate structures, and corresponding pH or temperature values and were sourced from UniProt and PubChem[6,31]. The pH and temperature datasets comprised 636 and 572 samples, respectively, and exhibited a wide range of values from 3 to 10.5 and 4 to 85 degrees (Supplementary Fig. 5a, b). To validate the effectiveness of the Revised UniKP, we conducted five-fold cross-validation on both datasets. Our results showed a strong correlation between predicted and experimentally

measured $k_{cat}$ values (Fig. 5b, c), with a PCC of 0.61 and 0.69 for pH and temperature, respectively. Moreover, the $R^2$ value was 0.36 and 0.47 for pH and temperature, respectively, indicating that the Revised UniKP performs well.

Furthermore, we divided the two datasets into training and test sets, allocating 80% and 20% respectively. Our results on independent test sets revealed that EF-UniKP outperforms both UniKP and Revised UniKP (Fig. 5d; $R^2 = 0.44$, $R^2 = 0.38$). On the pH dataset, the $R^2$ values for EF-UniKP were 20% and 8% higher than for UniKP and Revised UniKP, respectively. On the temperature dataset, the $R^2$ values were 26% and 2% higher, respectively. This was further supported by smaller errors and high correlation (Supplementary Fig. 6a, b). To prevent overfitting on the training set, we performed a stricter analysis and selected only those samples in the test set where at least either substrate or enzyme was not included in the training set, resulting in 62 and 61 samples for pH and temperature, respectively. Our results showed that EF-UniKP outperformed both UniKP and Revised UniKP, with $R^2$ values 13% and 10% higher on the pH dataset and 16% and 4% higher on the temperature dataset (Fig. 5e; $R^2 = 0.45$, $R^2 = 0.31$). The advantages in RMSE and PCC also confirmed the effectiveness of EF-UniKP (Supplementary Fig. 6c, d). Overall, our newly established pH and temperature dataset confirmed that our two-layer framework can effectively consider environmental factors and improve model performance.

**Enhancing high $k_{cat}$ prediction through re-weighting methods**

We further analyzed the $k_{cat}$ value distribution of the dataset used and found that it was highly imbalanced, with only a few samples at both ends and most samples in the middle, resembling a normal distribution (Fig. 6a). This presented a challenge for a machine learning model to extract the critical information[23]. However, predicting high $k_{cat}$ values has been essential in enzymology and synthetic biology[14,15]. To

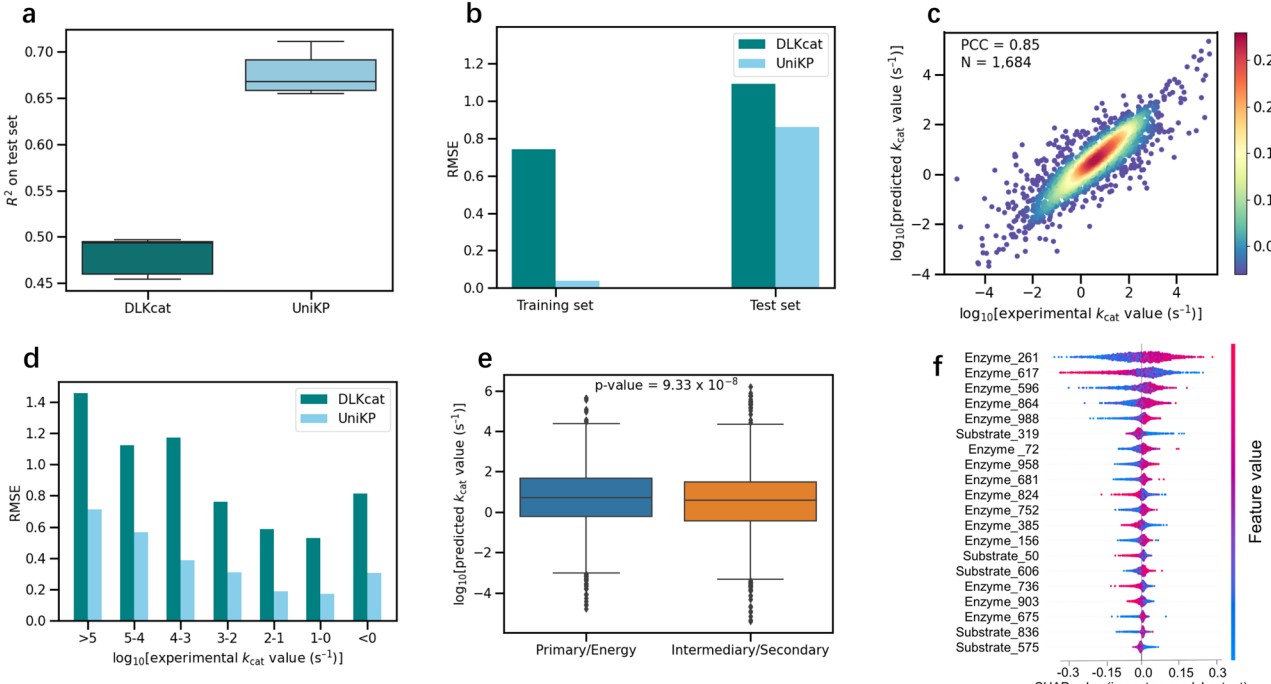

**Fig. 3 | High accuracy of UniKP in enzyme $k_{cat}$ prediction. a** Comparison of average coefficient of determination ($R^2$) values for DLKcat and UniKP after five rounds of random test set splitting ($n = 1684$). **b** Comparison of the root mean square error (RMSE) between experimentally measured $k_{cat}$ values and predicted $k_{cat}$ values of DLKcat and UniKP for training ($n = 15,154$) and test sets ($n = 1684$). Dark bars represent RMSE of DLKcat and light bars for UniKP. **c** Scatter plot illustrating the Pearson coefficient correlation (PCC) between experimentally measured $k_{cat}$ values and predicted $k_{cat}$ values of UniKP for the test set ($N = 1684$), showing a strong linear correlation. The color gradient represents the density of data points, ranging from blue (0.02) to red (0.28). **d** Comparison of RMSE between experimentally measured $k_{cat}$ values and predicted $k_{cat}$ values of DLKcat and UniKP in various experimental $k_{cat}$ numerical intervals. Dark bars represent RMSE of DLKcat and light bars for UniKP. **e** Enzymes with significantly different $k_{cat}$ values between primary central and energy metabolism, and intermediary and secondary metabolism. An independent two-sided $t$-test to determine whether the means of two independent samples differ significantly. Primary central and energy metabolism ($n = 3098$) and intermediary and secondary metabolism ($n = 4201$) were examined in this analysis. **f** Shapley additive explanations (SHAP) analysis for the top 20-feature Extra Trees model. The impact of each feature on $k_{cat}$ values is illustrated through a swarm plot of their corresponding SHAP values. The color of the dot represents the relative value of the feature in the dataset (high-to-low depicted as red-to-blue). The horizontal location of the dots shows whether the effect of that feature value contributed positively or negatively in that prediction instance (x-axis). In each box plot (**a, e**), the central band represents the median value, the box represents the upper and lower quartiles and the whiskers extend up to 1.5 times the interquartile range beyond the box range. Source data are provided as a Source Data file.

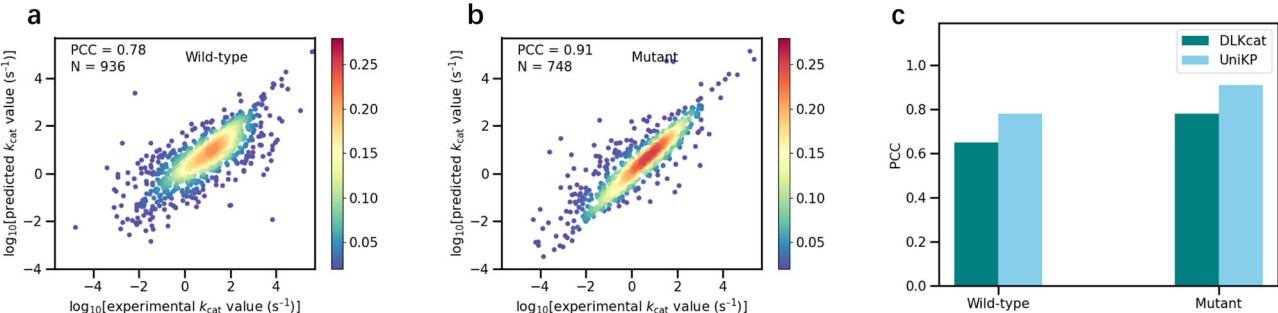

**Fig. 4 | UniKP markedly discriminates $k_{cat}$ values of enzymes and their mutants.** Scatter plot illustrating the Pearson coefficient correlation (PCC) between experimentally measured $k_{cat}$ values and predicted $k_{cat}$ values of UniKP for wild type enzymes (**a**) ($N = 936$) and mutated enzymes (**b**) ($N = 748$). The color gradient represents the density of data points, ranging from blue (0.02) to red (0.28). **c** PCC values of wild-type and mutated enzymes on the test set of DLKcat and UniKP. Dark bars represent PCC values of DLKcat and light bars for UniKP. Source data are provided as a Source Data file.

illustrate this, we implemented five-fold cross-validation on the entire dataset to independently scan the predicted $k_{cat}$ for all samples, from which we observed that the error was higher at both ends of the dataset compared to the middle (Fig. 6b). To address this issue, we utilized representative re-weighting methods, including Directly Modified sample Weight (DMW), Cost-Sensitive re-Weighting methods (CSW), Class-Balanced re-Weighting methods (CBW), and Label Distribution Smoothing (LDS), on the $k_{cat}$ dataset[24,32,33]. For each method,

we applied several commonly used hyperparameters to the entire dataset and performed five-fold cross-validation using UniKP to predict $k_{cat}$ values for all samples, whereas 149 samples with $k_{cat}$ greater than 4 were selected to represent high $k_{cat}$ samples for DMW optimization and downstream comparison (Supplementary Figs. 7–11). We found that all of these methods outperformed the initial UniKP, with CBW being the most effective. The RMSE of high $k_{cat}$ samples was 6.5% lower with CBW than with the initial model (Fig. 6c). We further

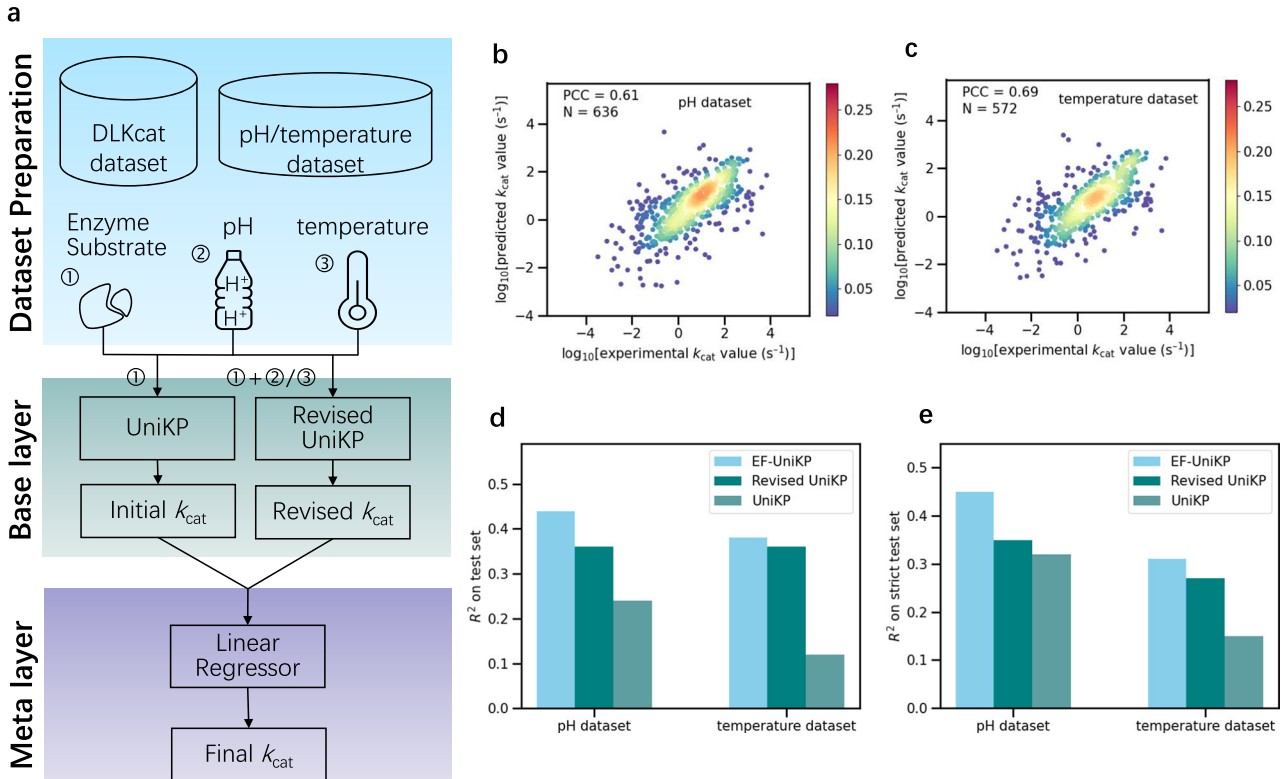

**Fig. 5 | A two-layer framework considering environmental factors. a** A two-layer framework called EF-UniKP that consists of a base layer and a meta layer. The base layer contains two models, namely UniKP and Revised UniKP. The UniKP takes the concatenated representation vector of the enzyme and substrate as input, while the Revised UniKP uses a concatenated representation vector of the enzyme and substrate, combined with the pH or temperature value. Both models are trained using the Extra Trees algorithm. The meta layer of this framework includes a linear regressor that uses the predicted $k_{cat}$ values from both the UniKP and Revised UniKP to predict the final $k_{cat}$ value. Scatter plot illustrating the Pearson coefficient correlation (PCC) between experimentally measured $k_{cat}$ values and predicted $k_{cat}$ values of Revised UniKP for pH set (**b**) ($N$ = 636) and temperature set (**c**) ($N$ = 572). The color gradient represents the density of data points, ranging from blue (0.02) to red (0.28). **d** Coefficient of determination ($R^2$) values between experimentally measured $k_{cat}$ values and predicted $k_{cat}$ values on pH and temperature test sets of EF-UniKP, Revised UniKP and UniKP. Light bars represent $R^2$ of EF-UniKP, dark bars for Revised UniKP and darkish bars for UniKP. **e** $R^2$ values between experimentally measured $k_{cat}$ values and predicted $k_{cat}$ values on more strict pH and temperature test sets of EF-UniKP, Revised UniKP and UniKP. These are the samples in the test set where at least either the substrate or enzyme was not included in the training set, resulting in 62 and 61 samples for pH and temperature, respectively. Light bars represent $R^2$ of EF-UniKP, dark bars for Revised UniKP and darkish bars for UniKP. Source data are provided as a Source Data file.

subdivided the high $k_{cat}$ value samples into two numerical intervals, higher than 5 and between 4 and 5, and found that the smaller errors in both intervals confirmed the effectiveness of our approach (Supplementary Fig. 12). Overall, our use of representative re-weighting methods enabled us to adjust the sample weight distribution and effectively improve high $k_{cat}$ value prediction.

## Unified framework for $K_m$ and $k_{cat}$ / $K_m$ predictions

Next, we investigated the performance of UniKP on $K_m$ and $k_{cat}$ / $K_m$ prediction. Although $k_{cat}$ is traditionally considered as independent of $K_m$, in light of the principle that the primary sequence of a specific protein determined its three-dimensional structure, and therefore its function, we believed that hidden information in the primary sequence could also be used to predict its $K_m$ and $k_{cat}$ / $K_m$ values. For the prediction of Michaelis constant $K_m$ values, a reported representative dataset was selected[10], consisting of 11,722 natural enzyme-substrate combinations and their corresponding $K_m$ values. The dataset was randomly divided into a training set (80%) and a test set (20%) which was used to train a predictor capable of predicting $K_m$ with enzyme sequence. The results showed that the $K_m$ predictor trained by UniKP outperformed most baseline models and achieved the prediction performance of the state-of-the-art model, with $R^2$ = 0.530 and a RMSE value smaller than or equal to the previous models on the test set (Fig. 6d, e). There was also a high correlation

between the predicted $K_m$ values and experimentally measured $K_m$ values (PCC = 0.73).

For $k_{cat}$ / $K_m$ prediction, an additional dataset was collected from BRENDA, UniProt and PubChem[6,7,31], containing 910 enzyme sequences, substrate structures, and their corresponding $k_{cat}$ / $K_m$ values. Five-fold cross-validation was performed and a high correlation was found between the predicted values and real values (Fig. 6f; PCC = 0.81). The $R^2$ was 0.65 and the RMSE was 1.07. As there was no predictor available to directly predict $k_{cat}$ / $K_m$ value from enzyme sequence and substrate structures, the state-of-the-art $k_{cat}$ predictor and $K_m$ predictor were used to predict the corresponding $k_{cat}$ and $K_m$ parameters which were used to calculate the $k_{cat}$ / $K_m$ values[10,12]. The PCC of these calculated $k_{cat}$ / $K_m$ values with the corresponding experimentally measured ones was −0.02, demonstrating the significant advantages of UniKP. The advantage of UniKP can be attributed to the high prediction consistency of its unified framework. In conclusion, the generalizability of UniKP on similar tasks with small molecule-protein interactions was verified.

## UniKP assisted enzyme mining and evolution

Discovering alternative enzymes with enhanced activity for specific biochemical reactions and improving known enzyme efficiency through directed evolution are vital in synthetic biology and biochemistry research. However, enzyme mining and evolution processes

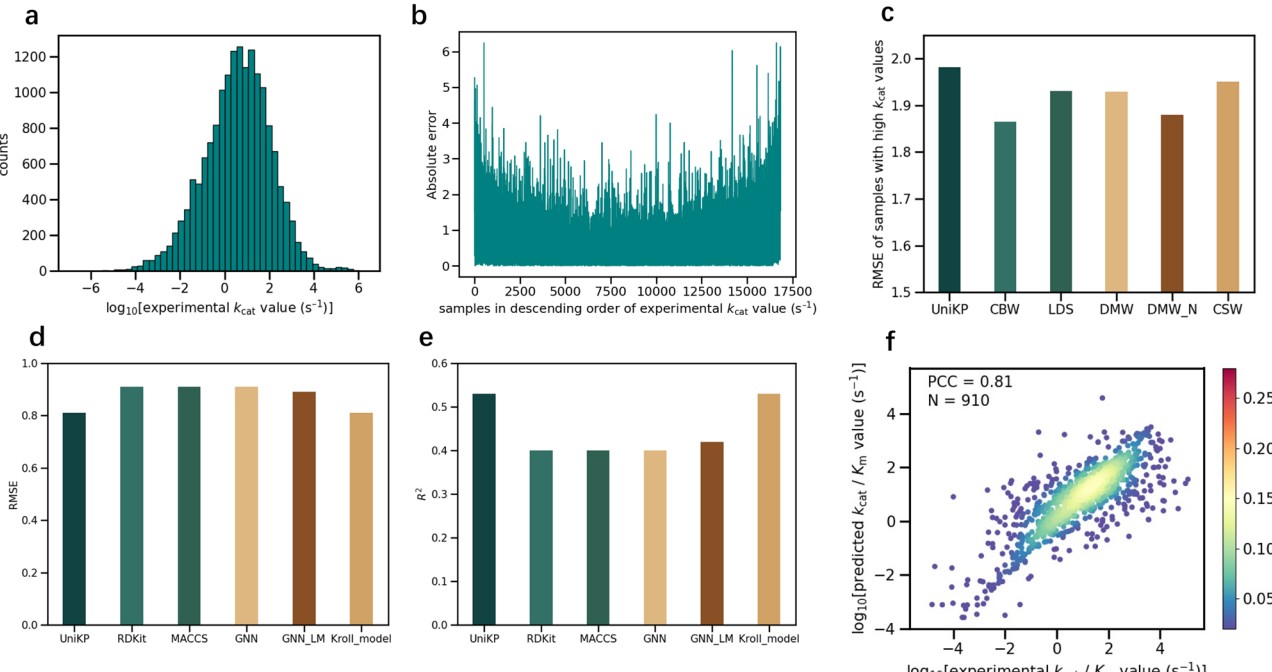

**Fig. 6 | Enhancing high $k_{cat}$ prediction through re-weighting methods and unified framework for $K_m$ and $k_{cat}$ / $K_m$ predictions. a** The distribution of $k_{cat}$ values in the $k_{cat}$ dataset. All samples are divided into 50 bins. **b** The absolute error between experimentally measured $k_{cat}$ values and predicted $k_{cat}$ values of each sample. The $k_{cat}$ values of all samples were predicted independently using five-fold cross-validation. **c** Root mean square error (RMSE) between experimentally measured $k_{cat}$ values and predicted $k_{cat}$ values of 149 samples with $k_{cat}$ values higher than 4 (logarithm value) using various re-weighting methods and the initial UniKP. **d**, **e** RMSE, coefficient of determination ($R^2$) between experimentally measured $K_m$ values and predicted $K_m$ values on $K_m$ test set. **f** Scatter plot illustrating the Pearson coefficient correlation (PCC) between experimentally measured $k_{cat}$ / $K_m$ values and predicted $k_{cat}$ / $K_m$ values of UniKP for $k_{cat}$ / $K_m$ dataset ($N = 910$). The color gradient represents the density of data points, ranging from blue (0.02) to red (0.28). Source data are provided as a Source Data file.

can be time-consuming and labour-intensive. Tyrosine ammonia lyase (TAL), a key rate-limiting enzyme in flavonoid biosynthesis, has been extensively studied to identify superior TAL variants. For example, one study selected 107 representative TALs from 4729 unique sequences found in over ten distinct microbial species through bioinformatic analysis[34]. However, enzymatic assays revealed that only four TALs displayed higher activities, with the highest $k_{cat}$ of 4.32 s⁻¹, considerably lower than the $k_{cat}$ of 114 s⁻¹ reported for a TAL from *Rhodotorula glutinis*, identified from bacterial and fungal TAL enzymes based on empirical knowledge[35–37]. Directed evolution of RgTAL resulted in only a slight improvement in its $k_{cat}$, from 114 to 142 s⁻¹, from laborious screening of 4800 mutants[36].

To employ UniKP for enzyme mining, we first used RgTAL as a template to perform BLASTp search and identified the top 1000 TAL sequences with high similarity. UniKP then predicted the $k_{cat}$ values for each sequence. The top 5 sequences with the highest predicted $k_{cat}$ values were selected for experimental validation. The results demonstrated that 5 RgTAL analogues can be successfully expressed in E. coli, of which 4 samples exhibited catalytic activity (Table 1, entry 1). Interestingly, 2 of the 4 samples surpassed RgTAL, with AsTAL from *Armillaria solidipes* displayed the highest $k_{cat}$ of 448.8 s⁻¹, four times greater than RgTAL.

To showcase UniKP's ability to assist directed evolution, we generated all variants with single-point RgTAL mutation, where each variant involved mutating an amino acid at a specific position to one of the other 19 canonical amino acids. This resulted in a total number of variants equal to the product of 19 and the length of the sequence (19*693 = 13,167). We performed an in-silico screening of all 13,167 single-point RgTAL mutations using UniKP by predicting their corresponding $k_{cat}$ and $k_{cat}$ / $K_m$ values. A total of 10 sequences, with the top 5 hits with highest predicted $k_{cat}$ and 5 with highest $k_{cat}$ / $K_m$, were chosen for experimental validation (Table 1, entry 2).

Among the mutants with highest predicted $k_{cat}$, one exhibited a slightly higher $k_{cat}$ / $K_m$ of 493.6 mM⁻¹s⁻¹, indicating increased efficiency compared to the wild-type enzyme. In contrast, among the mutants with highest predicted $k_{cat}$ / $K_m$, two mutants, RgTAL-10Y and RgTAL-489T, displayed significantly higher $k_{cat}$ / $K_m$ of 1079 mM⁻¹s⁻¹ and 1150 mM⁻¹s⁻¹, respectively. RgTAL-489T was 3.5-fold more efficient than the wild-type enzyme, representing the most efficient TAL reported to date. Moreover, through BLASTp analysis, the result showed that the most similar sequence in the whole dataset for $k_{cat}$ / $K_m$ predictor shares only a 35.42% identity. This demonstrates that UniKP indeed captures deep-level information, enabling effective screening of enzyme-substrate combinations that have not been presented in training set. These results confirm that UniKP is a rapid and effective method for mining new enzymes or improving existing enzymes for specific substrates.

Furthermore, in order to illustrate the effectiveness of EF-UniKP, we also conducted wet-lab experimental validations, using pH as an example. Specifically, we selected the TALclu[38], which exhibited an optimal catalytic pH of 9.5. We employed a similar enzyme mining approach as before against TALclu and selected the top 5 sequences with the highest predicted $k_{cat}$ values by EF-UniKP for experimental validation. Remarkably, we found that the $k_{cat}$ and $k_{cat}$ / $K_m$ values of all 5 sequences exceeded those of TALclu. The $k_{cat}$ value of HiTAL from *Heterobasidion irregulare* TC 32-1 was the highest at 76.00 s⁻¹, which is 4.6 times greater than that of TALclu. Additionally, the $k_{cat}$ / $K_m$ value of TrTAL from Tephrocybe rancida was the highest at 863.50 mM⁻¹s⁻¹, representing a 2.6-fold increase compared to TALclu (Table 1, entry 3). This result further demonstrates that EF-UniKP, when considering environmental factors, consistently identifies highly active TAL enzymes with remarkable precision.

## Table 1 | UniKP and EF-UniKP assisted enzyme mining and evolution

| Entry | Category | TALs | $k_{cat}$ (s$^{-1}$) | $K_m$ (mM) | $k_{cat} / K_m$ (s$^{-1}$·mM$^{-1}$) |
|---|---|---|---|---|---|
| 1 | UniKP: Enzyme mining | RgTAL | 117.8 | 0.36 | 327.2 |
| | | HiTAL | NA | NA | NA |
| | | PcTAL | 66.57 | 0.17 | 391.6 |
| | | SsTAL | 58.87 | 0.17 | 346.3 |
| | | AsTAL | 448.8 | 0.54 | 831.1 |
| | | IsTAL | 119.7 | 0.57 | 210.0 |
| 2 | UniKP: Enzyme evolution | RgTAL | 117.8 | 0.36 | 327.2 |
| | | MT-613P | NA | NA | NA |
| | | MT-603P | 52.57 | 1.15 | 45.71 |
| | | MT-366H | 40.31 | 0.48 | 83.98 |
| | | MT-366W | 31.02 | 0.85 | 36.49 |
| | | MT-587V | 162.9 | 0.33 | 493.6 |
| | | MT-10Y | 884.4 | 0.82 | 1079 |
| | | MT-337C | 96.81 | 6.48 | 14.94 |
| | | MT-668S | 53.60 | 3.85 | 13.92 |
| | | MT-489T | 816.8 | 0.71 | 1150 |
| | | MT-337D | 34.77 | 0.85 | 40.91 |
| 3 | EF-UniKP | TALclu | 16.54 | 0.05 | 330.80 |
| | | TrTAL | 34.54 | 0.04 | 863.50 |
| | | HiTAL | 76.00 | 0.21 | 361.90 |
| | | LeTAL | 33.85 | 0.07 | 483.57 |
| | | PpTAL | 28.09 | 0.04 | 702.25 |
| | | AaTAL | 25.24 | 0.03 | 841.33 |

**Entry 1)** The kinetic characteristics of wild-type Tyrosine ammonia lyase (RgTAL) from *Rhodotorula glutinis* and newly discovered TALs mined from non-redundant protein database by performing BLASTp. The top 5 sequences with the highest predicted $k_{cat}$ values by UniKP were selected for experimental validation, including HiTAL from *Heterobasidion irregulare TC 32-1* (XP_009553370.1), PcTAL from *Puccinia coronata f. sp. avenae* (PLW06342.1), SsTAL from *Sporidiobolus salmonicolor* (CEQ38810.1), AsTAL from *Armillaria solidipes* (BK74450.1), IsTAL from *Ilyonectria sp. MPI-CAGE-AT-0026* (KAH6995648.1). NA denotes the enzyme that was not soluble and showed no catalytic activity. **Entry 2)** The kinetic characteristics of RgTAL and mutants generated by UniKP. All variants of single-point mutations were generated for RgTAL, where each variant involved mutating an amino acid at a specific position to one of the other 19 canonical amino acids, which resulted in a total number of variants equal to the product of 19 and the length of the sequence (19*693 = 13,167). Through an in-silico screening of all 13,167 single-point mutations of RgTAL using UniKP, the top 5 mutants ranked by their predicted $k_{cat}$ or $k_{cat} / K_m$ values were chosen from each screening ($k_{cat}$ or $k_{cat} / K_m$) for experimental validation. NA denotes the enzyme that was not soluble and showed no catalytic activity. And MT denotes the mutated form of RgTAL. **Entry 3)** The kinetic characteristics of wild-type Tyrosine ammonia lyase from *Chryseobacterium luteum* sp. nov (TALclu) and newly discovered TALs mined from non-redundant protein database by performing BLASTp. The top 5 sequences with the highest predicted $k_{cat}$ values by UniKP were selected for experimental validation, including TrTAL from *Tephrocybe rancida* (KAG6920185.1), HiTAL from *Heterobasidion irregulare* TC 32-1 (XP_009553370.1), LeTAL from *Lentinula edodes* (KAF8828722.1), PpTAL from *Pleurotus pulmonarius* (KAF4563271.1), AaTAL from *Aspergillus arachidicola* (KAE8337485.1). All the experiments were conducted under a pH of 9.5.

## Discussion

The traditional method of measuring enzyme kinetic parameters of diverse enzymes and their substrates through labor-insentive and time-consuming experiments hampers the development of enzymology and synthetic biology applications. To address this challenge, we present a pretrained language model-based enzyme kinetic parameters prediction framework, UniKP, which improves the accuracy of predictions for three essential enzyme kinetic parameters, $k_{cat}$, $K_m$, and $k_{cat} / K_m$, using only the enzyme sequence and substrate structure. Here, we conducted a comprehensive comparison of 16 diverse machine learning models and 2 deep learning models on the machine learning module of UniKP, with the extra trees emerging as the best model. We speculated that tree-based ensemble models are better suited for this issue, with relatively small datasets (-10k) and high-

dimensional features (2048d). They utilized decision trees to efficiently break down high-dimensional data into smaller subsets, enabling more efficient feature selection and data segmentation. The combination of multiple decision trees reduced model variance, thereby balancing the instability of individual trees and decreasing the model's sensitivity to training data, contributing to improving the model's generalization abilities. The simpler models may be limited by insufficient fitting capabilities, while deep learning models rely on a large number of labelled samples, complicated network designs, and tedious parameter tuning.

UniKP demonstrated remarkable performance compared to the previous state-of-the-art model, DLKcat, in the $k_{cat}$ prediction task with an average coefficient of determination of 0.68, which was 20% higher. We speculated that pretrained models have greatly contributed to the performance of UniKP by creating an easily learnable representation of enzyme sequences and substrate structures using unsupervised information from the entire database[18]. Our analysis of model learning showed that protein information has a dominant effect, possibly due to the complexity of enzyme structure compared to substrate structure. Additionally, UniKP can effectively capture the small differences in $k_{cat}$ values between enzymes and their mutants, including experimentally measured cases, which is crucial for enzyme design and modification. And The disparity between the high identity region (>70 identity) and low identity region (<50 identity) of the $R^2$ of UniKP predicted values and $R^2$ of the gmean method underscores the adeptness of UniKP in extracting deeper interconnected information, thereby demonstrating a higher predictive accuracy in these tasks.

Moreover, the current models do not consider environmental factors, which is a critical limitation in simulating real experimental conditions. To address this issue, we propose a two-layer framework, EF-UniKP, which takes into account environmental factors. Based on two newly constructed datasets with pH and temperature information, respectively, EF-UniKP shows improved performance compared to the initial UniKP. To our knowledge, this is an accurate, high-throughput, organism-independent, and environment-dependent $k_{cat}$ prediction. Additionally, this approach has the potential to be extended to include other factors, such as cosubstrate and NaCl concentration[39,40]. However, existing models do not account for the interplay between these factors due to the lack of combined data. As experimental techniques advance, including biofoundry lab automation and continuous evolution methods[41,42], we anticipate a surge in enzyme kinetic data. This influx will not only enrich the field but also enhance the accuracy of prediction models. Additionally, due to the high imbalance in the $k_{cat}$ dataset, which results in a high error on high $k_{cat}$ value predictions, we systematically explored four representative re-weighting methods to mitigate this issue. The results showed that the hyperparameter settings for each method were critical in improving high $k_{cat}$ value prediction. Compared to the initial UniKP, all of the methods resulted in lower errors, with CBW being the optimal method. CBW argues that as the number of samples increases, the additional benefit of a newly added data point will diminish, indicating information overlap among the data[33]. Therefore, it further optimizes the weight by taking into account this issue. In the $k_{cat}$ dataset, enzymes with high homology, substrates with similar structures, and enzyme mutants contain overlapping information, which could explain why CBW is effective in this particular case. Overall, these findings demonstrate that re-weighting methods can assist biologists in improving specific value predictions they focus on.

Furthermore, we confirmed the strong generalizability of the current framework on Michaelis constant ($K_m$) prediction and $k_{cat} / K_m$ prediction. UniKP achieved state-of-the-art performance in predicting $K_m$ values and, even more impressively, outperformed the combined results of current state-of-the-art models in predicting $k_{cat} / K_m$ values. Furthermore, we validated the UniKP framework based on experimentally measured $k_{cat} / K_m$ values and $k_{cat} / K_m$ values calculated using

$k_{cat}$ and $K_m$ prediction models on $k_{cat}/K_m$ dataset. It is to be noted that the correlation observed between the values derived from UniKP $k_{cat}$ / UniKP $K_m$ and the experimental $k_{cat}/K_m$ is relatively low (PCC = −0.01). This discrepancy is likely attributable to the disparate datasets employed in constructing the respective models, necessitating the development of a distinct model for predicting $k_{cat}/K_m$ values. In the future, with the availability of a unified dataset encompassing both $k_{cat}$ and $K_m$ values, it is anticipated that the calculated outputs from the $k_{cat}$ and $K_m$ models would closely align with those generated by a dedicated model for $k_{cat}/K_m$.

The application of UniKP to Tyrosine ammonia lyase (TAL) enzyme mining and directed evolution demonstrated its potential to revolutionize synthetic biology and biochemistry research. This study showed that UniKP effectively identified high-activity TALs and rapidly improved catalytic efficiency of an existing TAL, with RgTAL-489T exhibiting a $k_{cat}/K_m$ value 3.5 times higher than the wild-type enzyme. Additionally, the derived framework EF-UniKP, when considering environmental factors, consistently identifies highly active TAL enzymes with remarkable precision, with TrTAL from Tephrocybe rancida exhibiting a $k_{cat}/K_m$ value 2.6 times higher than the wild-type enzyme. The result showed that the $k_{cat}$ and $k_{cat}/K_m$ values of all 5 sequences exceeded those of the wild-type enzyme. By expediting enzyme discovery and optimization processes, UniKP holds promise as a powerful tool for advancing biocatalysis, drug discovery, metabolic engineering, and other fields that rely on enzyme-catalyzed processes.

However, there are still some limitations to the current version of UniKP. For instance, while UniKP is capable of differentiating $k_{cat}$ values of experimentally measured enzymes and their variants, the predicted $k_{cat}$ values are not sufficiently accurate. This may be due to an insufficient dataset compared to the number of known protein sequences and substrate structures. Although re-weighting methods can somewhat relieve the prediction bias caused by the imbalanced $k_{cat}$ dataset (~6.5% improvement), more significant improvement may be achieved through the synthetic minority oversampling technique and other sample synthesis methods[43,44]. A central objective in synthetic biology is the development of a digital cell, poised to revolutionize our methods of studying biology. A critical prerequisite for this endeavour is the meticulous determination of enzymatic parameters for all enzymes within the pathway. Tools assisted by artificial intelligence illuminate this challenge, offering a high-throughput approach to predicting enzymatic kinetics. However, despite the reduced errors in UniKP predictors compared to earlier models, inaccuracies remain a significant hurdle in crafting a precise metabolic model. The inclusion of a growing number of experimentally determined $k_{cat}$ and $K_m$ values, sourced from cutting-edge high-throughput experimental techniques like those employed in modern biofoundries, can enhance model accuracy. Furthermore, we intend to incorporate state-of-the-art algorithms, such as transfer learning, reinforcement learning, and other small sample learning algorithms to effectively process imbalanced datasets[45,46]. Moreover, we aim to explore more applications, including enzyme evolution and global analysis of organisms.

## Methods

### Dataset source and preprocessing
To evaluate the UniKP framework, we selected several representative datasets and constructed several datasets to verify its accuracy.

DLKcat dataset. The DLKcat dataset was prepared as in the original publication[12]. Specifically, we began by utilizing the DLKcat dataset, which is the most comprehensive and representative dataset based on enzyme sequences and substrate structures from BRENDA and SABIO-RK databases. Initially, the dataset contained 17,010 unique samples, but we excluded samples with substrate simplified molecular-input line-entry system (SMILES) containing "." or $k_{cat}$ values less than or equal to 0, as per the DLKcat instruction. This resulted in 16,838 samples, which encompassed 7822 unique protein sequences

from 851 organisms and 2672 unique substrates. All $k_{cat}$ values were converted to a logarithmic scale. The dataset was divided into training and test sets, with a ratio of 90% and 10%, respectively, which was repeated five times to obtain 5 randomized datasets for downstream model training and test, keeping the same as in the previous publication.

pH and temperature datasets. To predict the influence of environmental factors to $k_{cat}$, we constructed two datasets that contain enzyme sequences, substrate structures, and their corresponding pH or temperature values. We obtained the enzyme sequences, substrate names, and pH or temperature values from the Uniprot database[6]. To obtain the corresponding substrate structure, we downloaded it from the PubChem database based on the substrate name and generated a SMILES representation via a python script[31]. The pH dataset comprised 636 samples, consisting of 261 unique enzyme sequences and 331 unique substrates, which resulted in 520 unique enzyme-substrate pairs. The pH values ranged from 3 to 10.5. The temperature dataset contained 572 samples, consisting of 243 unique enzyme sequences and 302 unique substrates, which resulted in 461 unique enzyme-substrate pairs. The temperature values ranged from 4 to 85 degrees. To evaluate the performance of UniKP on these datasets, we divided each dataset into an 80% training set and a 20% test set.

Michaelis constant ($K_m$) dataset. To assess the generalizability of UniKP on related tasks, we utilized a representative dataset obtained from a previous publication with SOTA results[10], which contains data retrieved from BRENDA. This dataset consists of 11,722 samples, comprising of enzyme sequences, substrate molecular fingerprints, and corresponding $K_m$ values. We converted the substrate structures into SMILES representations and $\log_{10}$-transformed all $K_m$ values. To evaluate the performance of UniKP on this dataset, we randomly divided the entire dataset into 80% training data and 20% test data, keeping the same as in the previous publication.

$k_{cat}/K_m$ dataset. We constructed an additional dataset using information sourced from the BRENDA, UniProt, and PubChem databases[6,7,31]. This dataset comprises 910 samples consisting of enzyme sequences, substrate structures, and their corresponding $k_{cat}/K_m$ values. We first obtained the UniProt ID of the enzyme and the name of the substrate along with their $k_{cat}/K_m$ values from the BRENDA database. Then, the corresponding enzyme sequences and substrate structures were obtained from the UniProt and PubChem databasesusing the UniProt ID and the name of the substrate, repsectively. We divided the entire dataset into five parts randomly to evaluate the performance of UniKP.

### Construction of UniKP
We implemented the UniKP framework using torch v. 1.10.1+cu113 and sklearn v. 0.24.2. UniKP consists of a representation module and a machine learning module. The representation module is responsible for generating effective representations of the enzyme sequences and substrate structures. We used the ProtT5-XL-UniRef50 protein language model, which has been shown to be effective in predicting peptide and protein function, to generate an embedded vector for the enzyme sequence[18]. Every amino acid was converted into a 1024-dimensional vector on the last hidden layer, and the resulting vectors were summed and averaged. The final enzyme representation was a 1024-dimensional vector. For the substrate structure, we generated a SMILES and used a pretrained SMILES transformer to create a 1024-dimensional vector by concatenating the mean and max pooling of the last layer and the first outputs of the last and penultimate layers[20]. The representation module converted the enzyme sequence or substrate structure into a numerical representation using an unsupervised learning process, making it easier for machine learning models to learn. The second module was an Extra Trees model, a machine learning method that can effectively capture the relationship between the concatenated representation vectors of the enzyme sequence and

substrate structure and the $k_{cat}$ value. All experiments were conducted in a Linux environment running Ubuntu 20.04.5 on a server with 64 cores and 4 NVIDIA GeForce RTX 3080 GPUs. We used a single core and GPU for training.

## Model setting

The 16 machine learning models includes Linear Regression, Ridge Regression, Lasso Regression, Bayesian Ridge Regression, Elastic Net Regression, Decision Tree, Support Vector Regression, K Neighbors Regressor, Random Forest, Gradient Boosting, Extra Trees, AdaBoost, Bagging, XGBoost (XGB), LightGBM (LGBM), MultiLayer Perceptron (MLP) Regressor. Here, the MLP Regressor was regarded as a traditional machine learning due to its shallow network design. We implemented all machine models using sklearn v. 1.1.1, utilizing default parameters, without additional optimization. The Convolutional Neural Network (CNN) architecture employed in this study comprises a 1D convolutional layer (conv1) with 16 output channels and a kernel size of 3 for feature extraction, followed by a max-pooling layer (pool) with a kernel size of 2 for downsampling. Then the model further includes two fully connected layers (fc1 and fc2), with fc1 having 16 * 1023 input features and 64 output features, and fc2 having 64 input features and 1 output feature. The architecture of Recurrent Neural Network (RNN) utilized in this study involves an RNN layer (rnn) with 2048 input features, 128 hidden units, and 1 layer. Following the RNN layer, there are two fully connected layers (fc1 and fc2). The first layer (fc1) has 128 input features and 64 output features, while the second layer (fc2) has 64 input features and 1 output feature. During the training process, deep learning models were optimized using an Adam optimizer with a learning rate of 0.0001, employing Mean Square Error as the loss function. The batch size was configured to be 8192. All deep learning models were implemented using Python 3.6.9 with pytorch 1.10.1+cu113.

## Construction of EF-UniKP

We developed a framework, called EF-UniKP, which takes into account environmental factors such as pH and temperature. This two-layer framework comprises a base layer with two individual models: UniKP and Revised UniKP. The UniKP takes as input a concatenated representation vector of the protein and substrate, while the Revised UniKP uses a concatenated representation vector of the protein and substrate, combined with the pH or temperature value. Both models were trained using the Extra Trees algorithm. The meta layer of the framework consists of a linear regression model that utilizes the predicted $k_{cat}$ values from both the UniKP and Revised UniKP as inputs. The pH and temperature datasets were divided into training and test sets, with the former being 80% of the dataset. The training set was further split into two subsets: the first training set was 80% of the training set (or 64% of the entire dataset) and the second training set was 20% of the training set (or 16% of the entire dataset). The training process involved two steps. In the first step, UniKP was trained using the DLKcat dataset without environmental factors, while Revised UniKP was trained using the first training set of pH or temperature dataset. In the second step, a linear regression model was trained using the second training set of pH or temperature dataset, and the outputs from both models in the first layer. The evaluation was performed using the test data of the pH or temperature dataset. As the model's performance may be influenced by different training and test set division, which were generated randomly, we have taken the precaution to average the results three times to mitigate this risk.

## Evaluation metrics

To evaluate the performance of our framework, we utilized various metrics to compare the predicted $k_{cat}$ value and experimentally measured $k_{cat}$. Our selected metrics included the coefficient of determination ($R^2$) in Eq. 1, the pearson correlation coefficient (PCC) in Eq. (2),

the root mean square error (RMSE) in Eq. (3), and the mean absolute error (MAE) in Eq. (4). These equations utilize variables such as $y_{ie}$ for the experimentally measured $k_{cat}$ value, $y_{ip}$ for the predicted $k_{cat}$ value, $\bar{y}_e$ for the average of the experimentally measured $k_{cat}$ values, $\bar{y}_p$ for the average of the predicted $k_{cat}$ values, and n for the number of samples (which depends on the size of the selected dataset). In this manuscript, we have presented various metrics in different sections for the comparison with existing models.

$$R^2 = 1 - \frac{\sum_{i=1}^{n}(y_{ie}-y_{ip})^2}{\sum_{i=1}^{n}(y_{ie}-\bar{y}_e)^2} \tag{1}$$

$$PCC = \frac{1}{n}\frac{\sum_{i=1}^{n}(y_{ie}-\bar{y}_e)(y_{ip}-\bar{y}_p)}{\sqrt{\sum_{i=1}^{n}(y_{ie}-\bar{y}_e)^2}\sqrt{\sum_{i=1}^{n}(y_{ip}-\bar{y}_p)^2}} \tag{2}$$

$$RMSE = \sqrt{\frac{\sum_{i=1}^{n}(y_{ie}-y_{ip})^2}{n}} \tag{3}$$

$$MAE = \frac{\sum_{i=1}^{n}|y_{ie}-y_{ip}|}{n} \tag{4}$$

## Feature importance analysis by SHAP

We utilized SHapley Additive exPlanations (SHAP), a unified framework for analyzing model interpretability, to compute the importance value of each feature[29]. The assigned SHAP value represents the significance of the feature, with higher values indicating greater importance. Moreover, SHAP can also indicate the positive or negative effects of features. This framework has been widely used to interpret the importance of various biological problems, including Type IV Secreted Effectors prediction and anticancer peptide prediction[47,48]. We applied SHAP on the $k_{cat}$ test set, which comprises 1684 samples, based on the trained UniKP. The SHAP summary produced by TreeExplainer displayed the magnitude, distribution, and direction of every feature effect. Each dot on the graph represents a dataset sample, and the x-axis position denotes the SHAP value, while the change in color represents different feature values. The implementation of SHAP was achieved through a freely available Python package.

## t-SNE visualization

To better understand the distribution of embedded enzyme and substrate representations and explore the necessity of using machine learning, we utilized t-distributed stochastic neighbor embedding (t-SNE) to visualize the embedded enzyme and substrate vectors[26]. This widely used method has been employed to analyze feature distributions in biological tasks, such as antimicrobial peptide recognition and protein subcellular location[49–51]. We calculated embedded vectors for all 16,838 samples, which were concatenated and inputed into the t-SNE algorithm, which transformed them into two-dimensional vectors. We used the default parameters for the algorithm and normalized the resulting numerical values of the two-dimensional vector for each sample to display, as shown in Eq. 5, where $y_{it}$ denotes the value of the $i_{th}$ projected vector and $y_t$ denotes all values of the projected vector.

$$y_{it} = \frac{y_{it}-\min(y_t)}{\max(y_t)-\min(y_t)} \tag{5}$$

## Sample weight redistribution methods

We explored four different methods to adjust the weight of the samples for accurate high $k_{cat}$ prediction. These representative weight redistribution methods included Directly Modified Sample Weight (DMW), Cost-Sensitive re-Weighting methods (CSW), Class-Balanced re-Weighting methods (CBW), and Label Distribution Smoothing (LDS)[24,32,33]. To enable fair comparison of the methods, we employed

5-fold cross-validation on the entire dataset to ensure that all samples could be predicted independently. We then divided the predicted $k_{cat}$ values into different intervals and calculated RMSE and MAE separately for $k_{cat}$ values higher than 4 (logarithm value) and $k_{cat}$ values higher than 5 (logarithm value).

**DMW method.** The DMW method is a weight redistribution approach where the weight of samples with $k_{cat}$ values higher than 4 (logarithm value) is directly enhanced. We explored several parameters, including weight multipliers (2, 5, 10, 20, 50, 100) and whether to normalize the weights. Through this process, we analyzed twelve optimized model combinations, which revealed that a weight coefficient of 10 without normalization was optimal. Increasing or decreasing the coefficient resulted in higher RMSE and MAE in predicting high $k_{cat}$ values.

**CSW method.** The CSW method assigns different weights to different classes to guide the model to pay more attention to minority categories. Three CSW variants, including CSW, root CSW, and square CSW, were applied to all samples. All the samples were divided into 131 bins, each covering an equal numeric interval. For CSW, the weight of each sample was set to the reciprocal of the sample size in each bin. The root CSW and square CSW methods reset the weight of each sample by its square root and its square, respectively. We found that the root CSW was the most effective method.

**CBW method.** The CBW method posits that the value of adding new data points will decrease as the size of the dataset increases. To reflect this, the effective number of samples can be calculated using Eq. (6), where $n$ is the number of samples and $\beta$ is a hyperparameter that ranges between 0 and 1. The weighting of each sample is then set to 1 divided by the effective number of samples. We evaluated the CBW method using different beta values (0.7, 0.75, 0.8, 0.85, 0.9, 0.99, 0.999, 0.9999) and found that the optimal value was 0.9, which resulted in the lowest prediction RMSE and MAE compared to other settings.

$$E_n = (1 - \beta^n)(1 - \beta) \tag{6}$$

**LDS methods.** LDS is a simple, effective, and interpretable algorithm for tackling the problem of unbalanced datasets that exploits the similarity of the nearby label space. It had been verified to be very effective in sections where only a few samples exist, and the predicted error would be reduced dramatically. LDS convolves a symmetric kernel with the empirical density distribution to generate a kernel-smoothed effective density distribution in Eq. 7, where $p(y)$ denotes the number of appearances of label $y$ in the training data, $\bar{p}(y')$ denotes the effective density of label $y'$, and $k(y,y')$ denotes the symmetric kernel. Similarly, we selected a Gaussian kernel and set various kernel sizes (3, 5, 7) and sigma values (1, 2). The optimal kernel size and sigma were 5 and 1, respectively.

$$\bar{p}(y') = \int k(y,y')p(y)dy \tag{7}$$

### Experimental validation of UniKP
We attempted to utilize UniKP to boost the enzyme mining process. Specifically, we selected a crucial enzyme in the naringenin synthetic pathway, tyrosine ammonia lyase (TAL).

**BLASTp.** Basic Local Alignment Search Tool protein (BLASTp) is a widely used bioinformatics tool for sequence similarity search[52]. It is a protein-protein BLAST algorithm that compares the query protein sequence against a non-redundant protein database and retrieves similar sequences based on their E-value, which estimates the probability of observing the alignment by chance. In this study, we utilized the BLASTp algorithm with RgTAL from *Rhodotorula glutinis*

(AGZ04575) as the template to identify sequences with high similarity to TAL, and subsequently selected the top 1000 sequences based on E value for $k_{cat}$ prediction using the UniKP. The parameters used were default, employing a BLOSUM62 scoring matrix, a word size of 5, and an expectation threshold of 0.05 for the setting.

**Experimental materials.** The plasmids and strains used in the experiment are detailed in Supplementary Tables 1–2. For strain maintenance, Luria-Bertani (LB) medium, which contains 10 g/L tryptone, 10 g/L NaCl, and 5 g/L yeast extract, was utilized. In order to produce naringenin in diverse strains, MOPS (3-(N-morpholino)propanesulfonic acid) medium was used. All of the chemicals used in the experiment were reagent grade and purchased from Sigma-Aldrich (St. Louis, MO, USA). NEBuilder® HiFi DNA Assembly Kit (E2621S) was purchased from NEB (Beverly, MA, USA) for the purpose of plasmid constructions.

**Determination of enzymatic kinetic parameters.** The predicted TALs were codon-optimized for expression in BL21(DE3)[36]. These TALs were respectively synthesized and cloned into the pET32a plasmid by Genewiz (Suzhou, China). The enzyme kinetic parameters of the different TAL enzymes were also evaluated as the following processes. Specifically, the candidate enzymes were tested in a 200 μL reaction volume with purified protein (1 μg), different concentrations of $L$-tyrosine, and Tris-HCl buffer (90 μL 50 mM pH 8.5). The mixture was incubated at 40 °C for 30 min and monitored for the appearance of coumaric acid at 315 nm[36]. One unit of enzyme activity was defined as 1 μM p-coumaric acid production in one minute. The kinetic curves of experimental results were included in Supplementary Fig. 13–16. The chemicals in this study were analytical reagent grade and purchased from Sigma-Aldrich (Steinheim, Germany).

**HPLC methods for naringenin detection.** HPLC methods for detecting naringenin involved the use of an Agilent 1260 HPLC system (Waldbronn, Germany) equipped with a diode array detector (DAD) 1260 model VL + (G7115A) and a C18 column (3 × 100 mm 2.7 μm). The detection was performed at 290 nm and 30 °C. A gradient elution condition was employed with the following steps: 10% to 40% acetonitrile/water (vol/vol) for 5 min, 40% acetonitrile (vol/vol) for 7 min, 40% to 95% acetonitrile (vol/vol) for 3 min, and 95% to 10% acetonitrile (vol/vol) for 3 min. The elution rate was 0.3 mL/min. Additionally, 0.3% acetic acid (vol/vol) was added to the mobile phases to facilitate the separation of naringenin.

### Statistics and reproducibility
The source of the dataset and the division criteria presented in this paper are detailed in each results section. Statistical analyses were performed using the packages in Python 3 (https://www.python.org/). No statistical method was used to predetermine sample size. We excluded samples with substrate simplified molecular-input line-entry system (SMILES) containing "." or $k_{cat}$ values less than or equal to 0, as per the DLKcat instruction[12]. All datasets were randomly split into training and test sets to ensure a fair comparison. And based on the goal of enzyme evolution, samples with the highest enzyme kinetic parameters predicted by UniKP were selected for experimental validation. We were not blinded to allocation during experiments and outcome assessment.

### Reporting summary
Further information on research design is available in the Nature Portfolio Reporting Summary linked to this article.

## Data availability
All relevant data supporting the key findings of this study are available within the article and its Supplementary Information files. All the data

analysed in this study is publicly available from either public databases, including BRENDA (https://www.brenda-enzymes.org/search_result.php?a=305), UniProt (https://www.uniprot.org/), PubChem (https://www.uniprot.org/) databases or supplementary datasets of referenced articles (https://github.com/SysBioChalmers/DLKcat, https://github.com/AlexanderKroll/KM_prediction). The data described in this manuscript are available for download at https://github.com/Luo-SynBioLab/UniKP. The data is also available on Zenodo: https://doi.org/10.5281/zenodo.10115498[53]. Source data are provided with this paper.

## Code availability

In order to facilitate additional utilization, we have made available all of the codes and thorough instructions in our GitHub repository located at https://github.com/Luo-SynBioLab/UniKP. Furthermore, a user-friendly example for predicting enzyme kinetic parameters has also been included in the repository. The code is also available on Zenodo: https://doi.org/10.5281/zenodo.10115498[53].

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

## Acknowledgements

We would like to acknowledge the support from National Key R&D Program of China (2018YFA0903200 to X.L.), National Natural Science Foundation of China (32071421 to X.L.), Guangdong Basic and Applied Basic Research Foundation (2021B1515020049 to X.L.), Shenzhen Science and Technology Program (ZDSYS20210623091810032 and JCYJ20220531100207017 to X.L. and H.D.), and Shenzhen Institute of Synthetic Biology Scientific Research Program (ZTXM20203001 to X.L. and H.Y.). We also want to thank Miss Z. Wei for the support in handling administrative affairs.

## Author contributions

H.Y. and X.L. conceived and designed the study, analyzed and interpreted the data, drafted and revised the manuscript. H.D. contributed to the conception and design of the study, wet-lab experiments and data analysis, and critical revisions of the manuscript. J.H. contributed to the wet-lab experiments and data analysis. J.D.K. revised the manuscript. All authors read and approved the final manuscript.

## Competing interests

X.L. has a financial interest in Demetrix and Synceres. J.D.K. has a financial interest in Amyris, Lygos, Demetrix, Napigen, Maple Bio, Apertor Labs, Zero Acre Farms, Berkeley Yeast, and Ansa Biotechnology. The remaining authors declare no competing interests.
