## [Peer Review File · Nature Communications]

UniKP: A unified framework for the prediction of enzyme kinetic parametersReviewer #1 (Remarks to the Author):

The authors proposed a unified framework for the prediction of enzyme kinetic parameters, UniKP, and expanded its application to include temperature and pH information. UniKP showed good performance in predicting k_{cat} , K_m , and k_{cat}/K_m , assisting in finding the most efficient tyrosine ammonia lyase. The results indicate that UniKP is a valuable tool for deciphering the mechanisms of enzyme kinetics and enables novel insights into enzyme engineering and their industrial applications. It deserved to be published after considering the following concerns:

Major:

1. The authors introduced that the current approaches have treated k_{cat} and K_m prediction as two separated questions (Line 60), and this seems to be a potential problem. In my opinion k_{cat} and K_m are different parameters so they are certainly separate questions. And I don't understand the necessity of constructing a model that predicts k_{cat}/K_m , especially the authors already have k_{cat} and K_m prediction models. How about the results of k_{cat}/K_m calculated by the models constructed in this work? If the result is not satisfactory, does it mean that the performance of k_{cat} and K_m prediction model are also insufficient?
2. The authors collected different dataset to train UniKP to predict different parameters, especially for K_m and k_{cat}/K_m prediction, the framework is exactly the same. While DLKcat is a k_{cat} prediction model, it uses the information of protein sequence and molecule structure as input, which is the same as UniKP, so DLKcat model can also be used to predict K_m and k_{cat}/K_m if trained by different datasets. The "unified framework" for UniKP seems not a primary advantage, instead the EF-UniKP model that considers environmental factors is worth more discussion, for example, its experimental validation and potential application.

Minor:

3. UniKP outperformed deep learning-based DLKcat for k_{cat} prediction, is it the pretrained representation or the Extra Tree model that plays the major role? How about pretrained representation + deep learning? It is recommended to do some ablation experiments to verify the superiority of the method.
4. The pretrained model and Extra Tree should be discussed in greater depth rather than merely listed as references in the Introduction section, since they are at the heart of UniKP.
5. The metrics R^2 and PCC are intertwined in the manuscript, although the equations are given, it is not clear when we should use R^2 or PCC. Are both required? Can we only use one of them?
6. Line301: how to screen and obtain the single-point RgTAL mutations?
7. Line 302: what does "each group" mean, top 5 hits were identified while 10 mutants were listed in figure 6b? more description should be given in the figure.
8. Line 303-306: this sentence is difficult to understand.
9. There are two reference sections in the manuscript, it would be better to integrate them.
10. It is strongly recommended that this manuscript be thoroughly proofread and edited, including some tense errors and confused expression.

Reviewer #2 (Remarks to the Author):

This study presents a unified framework, UniKP, for predicting enzyme kinetic parameters, specifically k_{cat} and K_m . The authors employ deep learning models for data mining and demonstrate the discovery of active TAL enzymes as a noteworthy application. However, while the use of pretrained protein language models and the SMILES Transformer is common in protein-ligand interaction modeling, the novelty of UniKP remains insufficient. Furthermore, concerns regarding data leakage and several other inquiries and doubts need to be addressed for a more comprehensive understanding of the research. Whether considering publication in this journal or elsewhere, it is recommended that the authors incorporate new data and discussions to address the following points:

1. The novelty of UniKP's k_{cat} prediction appears limited. The authors acknowledge the small size of the enzyme kinetic data and justify their use of pretrained protein language models (ProtT5-XL-UniRef50) and the SMILES Transformer for generating meaningful representations. However, it is

crucial to differentiate this work from existing models or methods, given that other studies, such as Kroll et al. (Nat Commun 14, 2787, 2023), have also utilized pretrained protein language models for similar purposes. Unique contributions of UniKP should be clearly elucidated.

2. The manuscript should provide more details about the UniKP model, including the hyperparameters employed during training (such as batch size, learning rate, optimizer, and parameter freezing) and specific information about the model setup (such as the size of the Extra Tree model).

3. Line 548 mentions "substrate structure" in relation to SMILES representation. SMILES representation does not provide a complete structural view, and consideration of 3D conformers could offer a more holistic representation.

4. The partitioning of the training and test datasets based on sequence similarity is not explicitly described. To address potential data leakage issues and ensure accurate model evaluation, the authors should conduct model training and testing on datasets separated by sequence identities. Given the limited innovation in deep learning models presented in this research, a careful evaluation of data leakage would be a valuable contribution to the field. Additionally, important metrics such as RMSE (Root Mean Square Error) should be listed in a well-formatted table rather than presented solely in bar plots.

5. The presence of RgTAL mutations in the training data should be clarified to avoid any potential data leakage concerns.

6. The generalization of the model trained on kcat prediction to the prediction of Km and kcat/Km requires thorough discussion. In enzymology, Kcat is traditionally considered independent of Km. Thus, an explanation is needed to address the unexpected finding and prevent any misinterpretation that may lead to misconceptions about enzyme catalysis.

7. A discrepancy is observed between line 831 and Figure 5e regarding the representation of Kcat/Km. The authors should rectify this inconsistency to ensure clarity.

8. In Figure 5, a scatter plot depicting the predicted kcat against the experimental Km would provide a clearer visualization. Moreover, it is important to note that RMSE and R2 (coefficient of determination) should not share the same Y-axis due to their different dimensions. RMSE measures the difference between model predictions and actual values and is expressed in the same units as the predicted and actual values. As UniKP predicts kcat in s⁻¹ and Km is typically measured in mM, RMSE lacks practical significance. On the other hand, R2 represents the proportion of variance in the dependent variable that can be predicted from the independent variable, and it is a dimensionless value between 0 and 1. Since R2 describes linear correlation, it may be more suitable to include Spearman correlation for assessing the model's performance.

9. In Figure 6a, AsTAL exhibits a higher kcat than RgTAL, while another TAL enzyme, TALclu, demonstrates a kcat/Km four times higher than that of RgTAL (ChemBioChem 2022, 23, e2022000). This raises the question of why RgTAL was chosen, as it does not exhibit optimal enzymatic activity. It is important to clarify whether UniKP is limited to redesigning enzymes with mid-range kcat values, while unable to further improve enzymes that already possess the highest catalytic activity.

10. Lastly, the current description of Kcat determination lacks specificity. Including figures depicting the kinetic data of Kcat would enhance the credibility of the study.

I hope incorporating the suggested revisions and addressing the concerns and inquiries mentioned above will significantly improve the manuscript.

Reviewer #3 (Remarks to the Author):

In the manuscript presented by Yu, et al., a computational framework for prediction of kinetic parameters of enzymes, namely, turnover numbers (kcat), Michaelis constants (Km) and enzyme efficiency (kcat/Km), UniKP, is presented. This framework is based on pre-trained language models that use protein sequence and substrate structures as input features. Additionally, a two-layer machine learning approach is used for accounting for pH and temperature variations in predicted parameters, and different approaches for improving prediction of high kcat values are tested. The authors ran extensive comparison on predictive performance of their model vs. the state-of-the-art tools for prediction of kcat (Li et al. 2022) and Km (Kroll et al. 2023) parameters, showing improvements in correlation coefficients and error metrics. However, the prediction errors displayed by UniKP are still above an order of magnitude in kinetic parameters, on average, which limits large-scale applicability for accurate modeling of cellular function, leaving room for further improvements.

Notably, the authors show how this tool can be used for mining sequences and selecting the optimal enzyme for a specific metabolic reaction (tyrosine ammonia liase, TAL), and predict single-point mutations that increase enzymatic performance significantly, achieving the highest reported enzyme efficiency for a TAL mutant.

The subject addressed by this study is of current high relevance in the fields of synthetic and systems biology, machine learning and metabolic engineering, therefore, the results presented here are of high relevance for a wide public. The use of pre-trained language models for prediction of enzyme parameters offers a new perspective on the use of machine learning methods for biological purposes, the use of this method for enzyme discovery and directed improvement is a major novel contribution to the aforementioned fields, opening possibilities for many applications.

I recommend the authors to address the following major and minor points in order to improve the quality of this manuscript and support their findings even further.

Major points:

- 1.- Most of the model performance evaluations in this manuscript focus on the comparison of correlation, determination coefficients and error metrics with the previously published DLKcat. Recently, a preprint highlighting the flaws in the selection and partitioning of the DLKcat dataset (<https://doi.org/10.1101/2023.02.06.526991>) has been published in bioRxiv. This study demonstrates that DLKcat predictions are accurate just for enzymes that are highly similar to those used in the training dataset. They show that a simple approach of approximating Kcats, by computing geometric means of kcats across the 3 top-similar enzymes (sequence similarity) in the training dataset is capable of yielding significantly better determination coefficients for predictions of the test dataset. I recommend the authors to also run a comparison of this type, as the predictions of DLKcat have already been found to be biased.
- 2.- Additionally, as major part of the results and discussion focus on comparison with DLKcat, the large error metrics displayed by uniKP are not discussed in detail. This is common in AI-ML literature, where performance is usually evaluated in terms of correlation or model fitness. In this case, the AI framework proved to be useful for directed enzyme improvement. Nonetheless, readily available kinetic parameters in a large-scale has been one of the most important bottlenecks for advancing on mechanistic understanding of cell behavior through modeling attempts. For this whole field of biological sciences, it is crucial to provide accurate parameters that enable appropriate quantification of cellular states and dynamics. AI-ML methods offer a great opportunity for addressing this long-standing issue, but more development is still needed. This manuscript would benefit from a broader and deeper discussion on the limitations of accurate quantitative prediction of kinetic parameters and their implications for bottom-up biological studies.
- 3.- The use of a machine learning model for Kcat prediction is justified in the first results subsection by showing that the concatenated vectors (combined representation of sequence and substrate) cannot differentiate between Kcat values of different orders of magnitude, by projecting such vectors using t-SNE analysis. In the methods section it is mentioned that such analysis was ran using just the default parameters. It is not mentioned which was the specific software

implementation used for this, and default parameters may differ from one to another. Moreover, t-SNE is a stochastic method that relies on two main parameters (perplexity and number of iterations). The choice of these parameters is crucial, as they may have drastic effects on the projection results. In order to strengthen this section, I recommend the authors to run a parametric analysis of the t-SNE projections to show the effect of the parameters in the grouping of the concatenated enzyme vectors. If Kcat values do not seem to group despite the chosen perplexity and number of iterations, then the implementation of uniKP will be better justified.

Minor points:

4.- Lines 47-48: The applications and implications of this study focus a lot on the use of this method for enhancing enzyme engineering, however, large-scale and accurate prediction of kinetic parameters could also majorly benefit systems biology, providing better parameters for quantitative modeling studies aiming to understand biological networks, specially metabolism. The manuscript could benefit from adding this point here.

5.- Lines 52-54: It is mentioned that, in comparison to the availability of millions of sequences, uniprot just offers around 2,000 kcat values. On the other hand, tenths of thousands of kinetic parameters are available in BRENDA and SABIO-RK bases, and in their latest versions these databases have integrated uniprot identifiers to their entries, facilitating a larger connection between measured parameters and protein sequences. Please mention this, or even add the corresponding numbers if possible, in order to clarify this better for the readers.

6.- Lines 69-70: "...often deviate from the ground truth". What is such truth regarding enzyme parameters, in vitro measurements? in vivo values? (probably impossible to characterize), predictions from basic principles using quantum chemistry? Even more, modern models of science and its methods of discovery do not aim to unveil "truths" from nature, but to create falsifiable hypotheses that can be empirically tested, and proof to offer coherent and viable explanations that expand our understanding of nature and our ways to modify it/interact with it. Please avoid the use of these categorical terms.

7.- Overall in the introduction and abstract sections prediction of kinetic parameters is mentioned repeatedly, but it is never said that this refers to prediction of in vitro measured parameters, please clarify this where at least once in these sections.

8.- Line 140: "further demonstrating the superiority of UniKP". The top used synonyms of the word superiority are dominance, excellence and perfection, which I doubt that is what the authors wanted to express. UniKP shows improved performance in comparison to other frameworks, major in some aspects, but as any approximation it is also limited in its prediction power, I recommend to avoid the use of this charged term here and in other parts of the text. Science is not a competition, but rather a cumulative enterprise of humanity for understanding nature.

9.- Line 154: "Theoretically, the former should exhibit higher values". This has been explored by many studies before. Please add references.

10.- Line 154: "Our results revealed", as mentioned above, many studies have explored this issue before, using different approaches, therefore, this manuscript is not revealing this but confirming or showing that results agree with what has been previously reported. Please modify accordingly.

11.- Lines 171-176: Could the authors add any error metrics here? (RMSE, or median absolute errors). Enzyme parameters are crucial for quantitative studies, hence, provide a measure of the associated errors in predictions will inform the community better.

12.- Line 182-183: Could the authors elaborate on why the selected enzyme-substrate pair is a crucial one and what is its relevance for the field?

13.- Line 236: "predicting high kcat values was essential", these predictions are still gonna be essential for the field, even after publication of this manuscript. These predictions are very

challenging and more accurate frameworks for prediction are likely to appear in the upcoming years, with further understanding and expansion of machine learning methods. Please change the word "was" to has been or something similar.

14.- Line 316: "prediction framework, UniKP, which can accurately predict three essential enzyme kinetic parameters". UniKP offers improvements in comparison with previous methods. Nevertheless, its predictions may differ from in vitro measurements even by an order of magnitude, which questions the use of the term "accurately predict" in absolute terms. It would be more fair to mention that it improves accuracy of predictions.

15.- Line 337: "As the learnable dataset expands". Here the authors acknowledge that the available measurements will expand, which I agree. It is even likely that future techniques will enable high-throughput characterization of enzymes. Nonetheless, in most of the text, the tone gives the idea that high-throughput prediction is the only active direction in this topic. Adding some nuances here and probably citing latest research on experimental methods and research directions would benefit the discussion.

Response to Reviewer's Comments:

We express our sincere gratitude to all reviewers for their thorough and insightful feedback, and for giving us an opportunity to revise our manuscript. In light of their comments, we have meticulously revised the manuscript and included additional data where necessary. Below, we address each of the comments in detail.

Reviewer comments

Reviewer #1 (Remarks to the Author):

The authors proposed a unified framework for the prediction of enzyme kinetic parameters, UniKP, and expanded its application to include temperature and pH information. UniKP showed good performance in predicting k_{cat} , K_m , and k_{cat}/K_m , assisting in finding the most efficient tyrosine ammonia lyase. The results indicate that UniKP is a valuable tool for deciphering the mechanisms of enzyme kinetics and enables novel insights into enzyme engineering and their industrial applications. It deserved to be published after considering the following concerns:

Response:

We appreciate the reviewer's thoughtful evaluation of our manuscript and the constructive comments about our work which helped us improve the quality of our manuscript. We have carefully revised the manuscript according to the comments.

Major:

1. The authors introduced that the current approaches have treated k_{cat} and K_m prediction as two separated questions (Line 60), and this seems to be a potential problem. In my opinion k_{cat} and K_m are different parameters so they are certainly separate questions.

Response:

Firstly, thanks for your valuable comment. The k_{cat} and K_m are indeed distinct parameters in enzymology with different biological relevance. However, their corresponding prediction tasks in the view of an algorithm have some similarity, both representing the prediction of a specific value with the corresponding protein sequence and substrate structure. These tasks have the potential to be predicted within a unified framework. This unified framework, once refined, contributes to improving performance across various tasks, as reported in similar reference [1]. We revised it in the introduction of our revised manuscript for better understanding, as shown below.

"Researchers have attempted to utilize computational methods to accelerate the process of enzyme kinetic parameters prediction, but current approaches have exclusively concentrated on addressing one of these issues, overlooking the similarity of both tasks in reflecting the relationship of protein sequences towards substrate structures."

And I don't understand the necessity of constructing a model that predicts k_{cat}/K_m , especially the authors already have k_{cat} and K_m prediction models. How about the results of k_{cat}/K_m calculated by the models constructed in this work? If the result is not satisfactory, does it mean that the performance of k_{cat} and K_m prediction model are also insufficient?

Moreover, regarding your question about the necessity of constructing a k_{cat} / K_m prediction model alongside existing k_{cat} and K_m models, we believe that the prediction of k_{cat} / K_m can offer additional insights. The reason is that machine learning models offer powerful tools for enzyme kinetic parameters prediction, but they are heavily reliant on the data they are trained on. In the case of k_{cat} and K_m prediction models, their corresponding training data came from different sources with different curation standards, which may result in inconsistency, posing challenges in achieving great performance when the predicted values were used for further calculation (PCC = -0.01 between experimentally measured k_{cat} / K_m values and k_{cat} / K_m values calculated using k_{cat} and K_m prediction models on k_{cat} / K_m dataset).

However, if the same set of data was used to train both models, the calculation should work. To demonstrate this, we retrieved 658 samples containing $k_{\text{cat}} / K_{\text{m}}$ values from the BRENDA database, all of which were presented in the k_{cat} dataset. The corresponding K_{m} value can be calculated. Subsequently, we performed 5-fold cross-validation to independently predict the values of k_{cat} and K_{m} for all samples, and then calculated the corresponding $k_{\text{cat}} / K_{\text{m}}$ values for each sample. We observed a higher correlation coefficient of 0.64 between the calculated $k_{\text{cat}} / K_{\text{m}}$ values and experimentally measured $k_{\text{cat}} / K_{\text{m}}$. This further confirms that our proposed UniKP framework can indeed ensure consistent predictions when the data is consistent. Although the calculation has some accuracy, the PCC is still lower than prediction with $k_{\text{cat}} / K_{\text{m}}$ dataset owing to its relatively limited sample size. Therefore, we have presented different prediction models for each enzyme kinetic parameters to provide more insights for various downstream tasks.

We hope this clarifies the description and rationale behind our construction and demonstrates its potential value in the field.

2. The authors collected different dataset to train UniKP to predict different parameters, especially for K_{m} and $k_{\text{cat}}/K_{\text{m}}$ prediction, the framework is exactly the same. While DLKcat is a k_{cat} prediction model, it uses the information of protein sequence and molecule structure as input, which is the same as UniKP, so DLKcat model can also be used to predict K_{m} and $k_{\text{cat}}/K_{\text{m}}$ if trained by different datasets. The “unified framework” for UniKP seems not a primary advantage, instead the EF-UniKP model that considers environmental factors is worth more discussion, for example, its experimental validation and potential application.

Response:

Firstly, thank you for pointing out that the DLKcat model exhibits resemblances to UniKP in terms of the input features. Our intention was to emphasize that these kinetic parameter prediction issues could be predicted within a unified framework with a robust and strong performance. Therefore, we have slightly modified some of our expressions to make the manuscript clearer.

1. We have deleted “the lack of a unified framework and” from the abstract.

2. We have changed “This discrepancy highlights the absence of a unified method for calculating or predicting $k_{\text{cat}} / K_{\text{m}}$, which is a crucial parameter reflecting catalytic efficiency.” to “This discrepancy highlights the importance of a demonstration of a unified method for calculating or predicting $k_{\text{cat}} / K_{\text{m}}$, which is a crucial parameter reflecting catalytic efficiency.”

Secondly, we fully agree with your point regarding the exploration and validation of EF-UniKP. To illustrate this, we conducted additional experimental validations, taking pH as an example. Specifically, we selected the Tyrosine ammonia lyase (TAL) enzyme, TALclu [2], which exhibited optimal catalytic pH of 9.5, and tried to identify mutants with higher performance at pH 9.5 with EF-UniKP. We performed a Blast search against the TALclu and then selected the top 1000 sequences with the highest similarity for prediction using EF-UniKP, with a pH setting of 9.5. From the predictions, we selected the top 5 sequences with the highest predicted k_{cat} values for wet-lab experimental validation. Remarkably, we observed that the k_{cat} and $k_{\text{cat}} / K_{\text{m}}$ values for all five selected sequences exceeded those of TALclu, with two of the sequences achieving $k_{\text{cat}} / K_{\text{m}}$ values 2.5-2.6 times higher than that of TALclu (Fig. R1). This result demonstrates that EF-UniKP, when considering environmental factors, consistently identifies highly active TAL enzymes with remarkable precision. It offers the opportunity to screen highly active enzymes across various environments, including extreme conditions, thereby facilitating industrial applications.

TALs	k_{cat} (s ⁻¹)	K_m (mM)	k_{cat} / K_m (s ⁻¹ ·mM ⁻¹)
TALclu	16.54	0.05	330.80
TrTAL	34.54	0.04	863.50
HiTAL	76.00	0.21	361.90
LeTAL	33.85	0.07	483.57
PpTAL	28.09	0.04	702.25
AaTAL	25.24	0.03	841.33

Fig. R1 The kinetic characteristics of wild-type Tyrosine ammonia lyase from *Chryseobacterium luteum* sp. nov (TALclu) and newly discovered TALs mined from non-redundant protein database by performing BLASTP. The top 5 sequences with the highest predicted k_{cat} values by UniKP were selected for experimental validation, including TrTAL from *Tephrocye rancida* (KAG6920185.1), HiTAL from *Heterobasidion irregulare* TC 32-1 (XP_009553370.1), LeTAL from *Lentinula edodes* (KAF8828722.1), PpTAL from *Pleurotus pulmonarius* (KAF4563271.1), AaTAL from *Aspergillus arachidicola* (KAE8337485.1). All the experiments were conducted under a pH of 9.5.

We have added Fig. R1 as new Fig. 7c in our manuscript.

We also have added a detailed description in the main text of the revised manuscript, as shown below. “Furthermore, in order to illustrate the effectiveness of EF-UniKP, we also conducted wet-lab experimental validations, using pH as an example. Specifically, we selected the Tyrosine ammonia lyase (TAL) enzyme, TALclu [38], which exhibited optimal catalytic pH of 9.5. We employed a similar enzyme mining approach as before against TALclu and selected the top 5 sequences with the highest predicted k_{cat} values by EF-UniKP for experimental validation. Remarkably, we found that the k_{cat} and k_{cat} / K_m values of all 5 sequences exceeded those of TALclu. The k_{cat} value of HiTAL from *Heterobasidion irregulare* TC 32-1 was the highest at 76.00 s⁻¹, which is 4.6 times greater than that of TALclu. Additionally, the k_{cat} / K_m value of TrTAL from *Tephrocye rancida* was the highest at 863.50 mM⁻¹s⁻¹, representing a 2.6-fold increase compared to TALclu (Fig. 6c). This result further demonstrates that EF-UniKP, when considering environmental factors, consistently identifies highly active Tyrosine ammonia lyase (TAL) enzymes with remarkable precision.”

Minor:

3. UniKP outperformed deep learning-based DLKcat for kcat prediction, is it the pretrained representation or the Extra Tree model that plays the major role? How about pretrained representation + deep learning? It is recommended to do some ablation experiments to verify the superiority of the method.

Response:

Thank you for your insightful suggestion. We have conducted a comprehensive comparison of 16 diverse machine learning models, including basic linear regression to complex ensemble models, as well as 2 representative deep learning models, the Convolutional Neural Network (CNN) and the Recurrent Neural Network (RNN). The results demonstrated the significant advantage of tree-based ensemble models, with the Extra Trees model, in particular, outperforming the others significantly, as illustrated below (Fig. R2). As DLKcat used other representation method and deep learning model, the results suggested that it is the combination of the pretrained representation and the extra trees model contributed to the improvement. We have also discussed the possible reasons below.

Performance Comparison of Different Models

Fig. R2 Performance Comparison of Different Models. a, b, c, d) Comparison of Root Mean Square Error (RMSE), Pearson Correlation Coefficient (PCC), Mean Absolute Error (MAE), and R^2 (Coefficient of Determination) values between experimentally measured k_{cat} values and predicted k_{cat} values of 16 diverse machine learning models and 2 deep learning models. The k_{cat} values of all samples were predicted independently using 5-fold cross-validation. Each bar in the graph represents the models' performance with respect to this metric. The "Extra Trees" model is highlighted in yellow, while other models are depicted in blue. The corresponding numerical values for each bar are provided on the right side.

Firstly, for the model training process, it's important to consider that the datasets are relatively small (~10k) and the features are high-dimensional (2048d). In this case, ensemble models generally prove to be more suitable and exhibit better prediction performance [3]. Simpler linear models exhibit lower fitting capability, while more complex neural networks require a large number of labeled data, potentially making them unsuitable for this problem.

Secondly, deep learning models, while capable of capturing complex patterns, are highly reliant on effective parameter tuning. Their inherent flexibility to adapt to training data can sometimes lead to overfitting, underscoring the importance of meticulous tuning to strike the delicate balance between model complexity and generalization.

Furthermore, the high dimensionality of the embedded vectors, resulting from the concatenation of pretrained language model outputs, might cause challenges for neural network. The curse of dimensionality, where the data points become sparse in a high-dimensional space, can hinder the learning process. Tree models, with their inherent ability to partition the high-dimensional space into manageable regions, excel in addressing such situations and can effectively mitigate this issue, facilitating more efficient learning [4].

We also added the test of different models in our manuscript, as shown below.

"For the machine learning module, in order to explore the performance of different models, we conducted a comprehensive comparison of 16 diverse machine learning models, including basic linear regression to complex ensemble models, as well as 2 representative deep learning models, the

Convolutional Neural Network (CNN) and the Recurrent Neural Network (RNN) (Fig. 2). Overall, the results showed that simpler models like linear regression displayed relatively poor prediction performance (Linear Regression $R^2 = 0.38$). In contrast, ensemble models demonstrated better performance. Notably, random forests and extra trees significantly outperformed other models, with extra trees exhibiting the highest performance (Extra Trees $R^2 = 0.65$). However, both categories of deep learning models, due to their demanding requirements for intricate network design and fine-tuning, did not perform as effectively in comparison (CNN $R^2 = 0.10$, RNN $R^2 = 0.19$). The results confirmed the significant advantage of ensemble models, with the extra trees model standing out as the best model. It's worth noting that the datasets are relatively small (~10k) and the features are high-dimensional (2048d), making ensemble models more suitable. Simpler linear models exhibit lower fitting capability, while more complex neural networks require a large amount of labeled data, potentially making them unsuitable for this problem. Therefore, the concatenated representation vectors of both the protein and substrate are subsequently input into the interpretable extra trees model for the prediction of three distinct enzyme kinetic parameters (Fig. 1c)."

We have also expanded our discussion session as shown in next point.

4. The pretrained model and Extra Tree should be discussed in greater depth rather than merely listed as references in the Introduction section, since they are at the heart of UniKP.

Response:

We have added a detailed study of pretrained model with different models in the Results section as shown in the previous points, and a more detailed discussion in the section, as follows.

"Here, we conducted a comprehensive comparison of 16 diverse machine learning models and 2 deep learning models on the machine learning module of UniKP, with extra trees emerging as the best model. We speculated that tree-based ensemble models are better suited for this issue, with relatively small datasets (~10k) and high-dimensional features (2048d). They utilized decision trees to efficiently break down high-dimensional data into smaller subsets, enabling more efficient feature selection and data segmentation. And the combination of multiple decision trees reduced model variance, thereby balancing the instability of individual trees and decreasing the model's sensitivity to training data, contributing to improving the model's generalization abilities. The simpler models may be limited by insufficient fitting capabilities, while deep learning models rely on a large number of labelled samples, complicated network designs, and tedious parameter tuning."

5. The metrics R2 and PCC are intertwined in the manuscript, although the equations are given, it is not clear when we should use R2 or PCC. Are both required? Can we only use one of them?

Response:

We apologize for any confusion caused by the intertwined use of the R^2 and PCC metrics in the manuscript. They convey distinct meanings: R^2 measures the proportion of the variance in the dependent variable that can be explained by the independent variable, indicating the goodness of fit of a regression model. PCC quantifies the linear relationship between two continuous variables. In our manuscript, we have included both the R^2 and PCC metrics for all the data we generated. However, during the comparison with existing models, we only wrote the corresponding metric as we can find in the original literatures, which may be seemed like the intertwined use of these two different parameters.

6. Line301: how to screen and obtain the single-point RgTAL mutations?

Response:

We apologize for this unclear description. We have added a detailed explanation in manuscript, as shown below.

"To showcase UniKP's ability to assist directed evolution, we generated all variants with single-point RgTAL mutation, where each variant involved mutating an amino acid at a specific position to one of

the other 19 canonical amino acids. This resulted in a total number of variants equal to the product of 19 and the length of the sequence ($19 \times 693 = 13,167$).

7. Line 302: what does “each group” mean, top 5 hits were identified while 10 mutants were listed in figure 6b? more description should be given in the figure.

Response:

We apologize for any confusion caused in line 302. We have revised our manuscript, as follows.

“A total of 10 sequences, with the top 5 hits with highest predicted k_{cat} and 5 with highest predicted k_{cat} / K_m , were chosen for experimental validation (Fig. 6b).”

We also added a more detailed description in Figure 6b, as follows.

“b) The kinetic characteristics of RgTAL and mutants generated by UniKP. All variants of single-point mutations were generated for RgTAL, where each variant involved mutating an amino acid at a specific position to one of the other 19 canonical amino acids, which resulted in a total number of variants equal to the product of 19 and the length of the sequence ($19 \times 693 = 13,167$). Through an *in-silico* screening of all 13,167 single-point mutations of RgTAL using UniKP, the top 5 mutants ranked by their predicted k_{cat} or k_{cat} / K_m values were chosen from each screening (k_{cat} or k_{cat} / K_m) for experimental validation. NA denotes the enzyme that was not soluble and showed no catalytic activity. And MT denotes the mutated form of RgTAL.”

8. Line 303-306: this sentence is difficult to understand.

Response:

We are sorry for this unclear description and have revised this sentence, as shown below.

“Among the mutants with highest predicted k_{cat} , one exhibited a slightly higher k_{cat} / K_m of $493.6 \text{ mM}^{-1} \text{ s}^{-1}$, indicating increased efficiency compared to the wild-type enzyme. In contrast, among the mutants with highest predicted k_{cat} / K_m , two mutants, RgTAL-10Y and RgTAL-489T, displayed significant higher k_{cat} / K_m of $1079 \text{ mM}^{-1} \text{ s}^{-1}$ and $1150 \text{ mM}^{-1} \text{ s}^{-1}$, respectively.”

9. There are two reference sections in the manuscript, it would be better to integrate them.

Response:

We apologize for any confusion. We have now merged the two reference sections into one.

10. It is strongly recommended that this manuscript be thoroughly proofread and edited, including some tense errors and confused expression.

Response:

We appreciate your suggestion and have certainly taken steps to thoroughly proofread and edit the manuscript.

References:

- [1] Lei, Yipin, et al. "A deep-learning framework for multi-level peptide–protein interaction prediction." *Nature communications* 12.1 (2021): 5465.
- [2] Brack, Yannik, et al. "Discovery of Novel Tyrosine Ammonia Lyases for the Enzymatic Synthesis of p - Coumaric Acid." *ChemBioChem* 23.10 (2022): e202200062.
- [3] Yu, Han, et al. "IPPF-FE: an integrated peptide and protein function prediction framework based on fused features and ensemble models." *Briefings in Bioinformatics* 24.1 (2023): bbac476.
- [4] Xia, Bin, et al. "PETs: a stable and accurate predictor of protein-protein interacting sites based on extremely-randomized trees." *IEEE transactions on nanobioscience* 14.8 (2015): 882-893.

Reviewer #2 (Remarks to the Author):

This study presents a unified framework, UniKP, for predicting enzyme kinetic parameters, specifically k_{cat} and K_m . The authors employ deep learning models for data mining and demonstrate the discovery of active TAL enzymes as a noteworthy application. However, while the use of pretrained protein language models and the SMILES Transformer is common in protein-ligand interaction modeling, the novelty of UniKP remains insufficient. Furthermore, concerns regarding data leakage and several other inquiries and doubts need to be addressed for a more comprehensive understanding of the research. Whether considering publication in this journal or elsewhere, it is recommended that the authors incorporate new data and discussions to address the following points:

Response:

We appreciate the reviewer for the constructive comments about our work which helped us improve the quality of our manuscript. We have added new data, analysis and discussions and carefully revised the manuscript according to the comments.

1. The novelty of UniKP's k_{cat} prediction appears limited. The authors acknowledge the small size of the enzyme kinetic data and justify their use of pretrained protein language models (ProtT5-XL-UniRef50) and the SMILES Transformer for generating meaningful representations. However, it is crucial to differentiate this work from existing models or methods, given that other studies, such as Kroll et al. (Nat Commun 14, 2787, 2023), have also utilized pretrained protein language models for similar purposes. Unique contributions of UniKP should be clearly elucidated.

Response:

We genuinely appreciate your thoughtful insights and concerns. We would like to elucidate the major contributions of our work in the following aspects.

1. For the construction of UniKP, our work proposes a new framework called UniKP based on pretrained language models for the prediction of three categories of enzyme kinetic parameters, including k_{cat} , K_m , k_{cat} / K_m predictions.

2. We proposed a two layer framework called EF-UniKP to consider environmental factors and confirm its effectiveness in k_{cat} predictions under different pH and temperature using two new datasets specifically collected for this study.

3. We analyzed the k_{cat} value distribution of the dataset used and found that it was highly imbalanced, which poses a challenge for a machine learning model to extract the critical information. Therefore, we systematically explored four representative re-weighting methods to successfully reduce the prediction error in high k_{cat} values prediction tasks.

4. We performed comprehensive wet-lab experimental validation based on Tyrosine ammonia lyase (TAL), a key rate-limiting enzyme in flavonoid biosynthesis. Our findings illustrate that UniKP has the capability to mine new enzymes as well as evolve enzymes to achieve higher activity for a specific substrate, reducing time and cost for both enzyme mining and directed evolution.

Beyond this, we agreed with your suggestion and the suggestions from other reviewers, we have added the following two experiments to further demonstrate the contribution and advantage of our methods:

First, we have conducted a comprehensive comparison of 16 diverse machine learning models and 2 deep learning models and added a detailed analysis of these models to provide deeper insights into the learning process. We have also added a more detailed discussion on the choice of features versus models in the Discussion section, as follows.

“For the machine learning module, in order to explore the performance of different models, we conducted a comprehensive comparison of 16 diverse machine learning models, including basic linear regression to complex ensemble models, as well as 2 representative deep learning models, the Convolutional Neural Network (CNN) and the Recurrent Neural Network (RNN) (Fig. 2). Overall, the results showed that simpler models like linear regression displayed relatively poor prediction performance (Linear Regression $R^2 = 0.38$). In contrast, ensemble models demonstrated superior performance. Notably, random forests and extra trees significantly outperformed other models, with extra trees exhibiting the highest performance (Extra Trees $R^2 = 0.65$). However, both categories of deep learning models, due to their demanding requirements for intricate network design and fine-tuning, did not perform as effectively in comparison (CNN $R^2 = 0.10$, RNN $R^2 = 0.19$). The results confirmed the significant advantage of ensemble models, with the extra trees model standing out as the best model. It's worth noting that the datasets are relatively small (~10k) and the features are high-dimensional (2048d), making ensemble models more suitable. Simpler linear models exhibit lower fitting capability, while more complex neural networks require a large amount of labelled data, potentially making them unsuitable for this problem. Therefore, the concatenated representation vectors of both the protein and substrate are subsequently input into the interpretable extra trees model for the prediction of three distinct enzyme kinetic parameters (Fig. 1c).”

“Here, we conducted a comprehensive comparison of 16 diverse machine learning models and 2 deep learning models on the machine learning module of UniKP, with extra trees emerging as the best model. We speculated that tree-based ensemble models are better suited for this issue, with relatively small datasets (~10k) and high-dimensional features (2048d). They utilized decision trees to efficiently break down high-dimensional data into smaller subsets, enabling more efficient feature selection and data segmentation. And the combination of multiple decision trees reduced model variance, thereby balancing the instability of individual trees and decreasing the model's sensitivity to training data, contributing to improving the model's generalization abilities. The simpler models may be limited by insufficient fitting capabilities, while deep learning models rely on a large number of labelled samples, complicated network designs, and tedious parameter tuning.”

Second, we have further explored and validated the usefulness of EF-UniKP with additional wet-lab experiments. We have added a detailed description in our manuscript, as shown below.

“Furthermore, in order to illustrate the effectiveness of EF-UniKP, we also conducted wet-lab experimental validations, using pH as an example. Specifically, we selected the Tyrosine ammonia lyase (TAL) enzyme, TALclu [38], which exhibited optimal catalytic pH of 9.5. We employed a similar enzyme mining approach as before against TALclu and selected the top 5 sequences with the highest predicted k_{cat} values by EF-UniKP for experimental validation. Remarkably, we found that the k_{cat} and k_{cat} / K_m values of all 5 sequences exceeded those of TALclu. The k_{cat} value of HiTAL from *Heterobasidion irregulare* TC 32-1 was the highest at 76.00 s^{-1} , which is 4.6 times greater than that of TALclu. Additionally, the k_{cat} / K_m value of TrTAL from *Tephroclybe rancida* was the highest at $863.50 \text{ mM}^{-1}\text{s}^{-1}$, representing a 2.6-fold increase compared to TALclu (Fig. 6c). This result further demonstrates that EF-UniKP, when considering environmental factors, consistently identifies highly active Tyrosine ammonia lyase (TAL) enzymes with remarkable precision.”

We believed that these additional experiments demonstrated novel insight for the choice of suitable feature representations and machine learning models for relevant studies in the future, as well as provided guidance to the protein evolution processes. Therefore, UniKP has its unique contribution and potential values in related field.

2. The manuscript should provide more details about the UniKP model, including the hyperparameters employed during training (such as batch size, learning rate, optimizer, and parameter freezing) and specific information about the model setup (such as the size of the Extra Trees model).

Response:

We apologize for any confusion about the model implementation. Since we added the content for the model comparison, we have included specific details in Methods, as shown below.

“Model setting. The 16 machine learning models includes Linear Regression, Ridge Regression, Lasso Regression, Bayesian Ridge Regression, Elastic Net Regression, Decision Tree, Support Vector Regression, K Neighbors Regressor, Random Forest, Gradient Boosting, Extra Trees, AdaBoost, Bagging, XGBoost (XGB), LightGBM (LGBM), MultiLayer Perceptron (MLP) Regressor. Here, the MLP Regressor was regarded as a traditional machine learning due to its shallow network design. We implemented all machine models using sklearn v. 1.1.1, utilizing default parameters, without additional optimization. The Convolutional Neural Network (CNN) architecture employed in this study comprises a 1D convolutional layer (conv1) with 16 output channels and a kernel size of 3 for feature extraction, followed by a max-pooling layer (pool) with a kernel size of 2 for downsampling. Then the model further includes two fully connected layers (fc1 and fc2), with fc1 having 16 * 1023 input features and 64 output features, and fc2 having 64 input features and 1 output feature. The architecture of Recurrent Neural Network (RNN) utilized in this study involves an RNN layer (rnn) with 2048 input features, 128 hidden units, and 1 layer. Following the RNN layer, there are two fully connected layers (fc1 and fc2). The first layer (fc1) has 128 input features and 64 output features, while the second layer (fc2) has 64 input features and 1 output feature. During the training process, deep learning models were optimized using an Adam optimizer with a learning rate of 0.0001, employing Mean Square Error as the loss function. The batch size was configured to be 8,192. All deep learning models were implemented using Python 3.6.9 with pytorch 1.10.1+cu113.”

3. Line 548 mentions "substrate structure" in relation to SMILES representation. SMILES representation does not provide a complete structural view, and consideration of 3D conformers could offer a more holistic representation.

Response:

We apologize for any confusion caused by unclear description. Our intention was to introduce that we can obtain the corresponding SMILES representation from the substrate's structure and then input it into the subsequent model. In fact, beside the 3D conformation, there are also additional features of a molecule which can be incorporated in the representation processes, such as the surface charge distribution, van der waals surface area, etc. As demonstrated in the literature by our group [1] and others, this additional information sometimes provide additional accuracy for some specific questions, while they showed no effect in other applications. We believed this discrepancy emerged from the intrinsic differences in different biological applications. For example, whereas ionic interaction is important in enzyme-ligand interaction, the major contributor in protein folding is van der waals interaction (hydrophobic interaction). Therefore, it is important to test for the best representation for each application.

4. The partitioning of the training and test datasets based on sequence similarity is not explicitly described. To address potential data leakage issues and ensure accurate model evaluation, the authors should conduct model training and testing on datasets separated by sequence identities. Given the limited innovation in deep learning models presented in this research, a careful evaluation of data leakage would be a valuable contribution to the field. Additionally, important metrics such as RMSE (Root Mean Square Error) should be listed in a well-formatted table rather than presented solely in bar plots.

Response:

We appreciate your valuable suggestion. In order to evaluate the potential data leakage, we used a method as reported in a recent bioRxiv manuscript [2], where the R^2 of geometric mean of the k_{cat} values of the three most similar enzymes for each enzyme in the training dataset (represent the potential leaking information) was compared with corresponding UniKP predicted values. The result demonstrated a clear advantage (higher correlation) of UniKP over the geometric means, as well as over DLKcat. Similarly, UniKP also exhibited a lower RMSE compared to the potential data leakage (if any), as shown in Fig R1.

Fig. R1 Performance Comparison of UniKP and geometric mean (gmean). a, b) The curves are coefficients of variation R^2 , RMSE, calculated in sliding windows of size $n=100$ across sequences in the test set ordered by the maximal sequence identity between individual test enzymes and all sequences in the training data. Position on the x-axis indicates the mean across the window. Red: UniKP predictions; yellow: geometric mean of k_{cat} values, calculated over the three most similar enzymes in the training set. The two points at the top right are for test datapoints with enzymes already used for training (100% max. sequence identity), these were not included in the sliding windows.

We have also added the following sentences in the Results section of the revised manuscript:

“In order to evaluate the potential data leakage, we used a reported method to compare the performance of UniKP with the geometric mean of experimental data which represent the potential data leakage [30]. The result demonstrated a clear advantage (higher correlation and lower RMSE) of UniKP over the geometric means, indicating the absence of data leakage in the training process (Supplementary Fig. 3).”

For the presentation of RMSE, to enable a more straightforward comparison with DLKcat, we adopted a similar approach as outlined in DLKcat or the paper you mentioned (Nat Commun 14, 2787, 2023).

We conducted a direct comparison using bar plots, and all numerical values are available in the source data for reference.

5. The presence of RgTAL mutations in the training data should be clarified to avoid any potential data leakage concerns.

Response:

We appreciate your careful consideration of data leakage concern. The variant with the highest k_{cat} / K_m value is RgTAL-489T, which is 3.5 times higher than RgTAL, as predicted by the k_{cat} / K_m predictor. Through BLAST analysis, the result showed that the most similar sequence in the whole dataset for k_{cat} / K_m predictor shares only a 35.42% identity. This demonstrates that UniKP indeed captures deep-level information, enabling effective screening of enzyme-substrate combinations that have not been presented in training set.

We have also added the following sentences in the Results section of the revised manuscript:

“Moreover, through BLASTp analysis, the result showed that the most similar sequence in the whole dataset for k_{cat} / K_m predictor shares only a 35.42% identity. This demonstrates that UniKP indeed captures deep-level information, enabling effective screening of enzyme-substrate combinations that have not been presented in training set.”

6. The generalization of the model trained on k_{cat} prediction to the prediction of K_m and k_{cat}/K_m requires thorough discussion. In enzymology, K_{cat} is traditionally considered independent of K_m . Thus, an explanation is needed to address the unexpected finding and prevent any misinterpretation that may lead to misconceptions about enzyme catalysis.

Response:

We apologize for any misunderstanding caused by my unclear explanation. Here, we were utilizing the same framework to train three distinct models based on three separate datasets. These models are entirely independent of each other. We have revised it in our manuscript for better understanding, as shown below.

1. We added the following sentences in the sub-section “Unified framework for K_m and k_{cat} / K_m predictions”: Although k_{cat} is traditionally considered as independent of K_m , in light of the principle that the primary sequence of a specific protein determined its three-dimensional structure, and therefore its function, we believed that hidden information in the primary sequence could also be used to predict its K_m and k_{cat} / K_m values.

2. We revised the following sentences in the introduction “Researchers have attempted to utilize computational methods to accelerate the process of enzyme kinetic parameters prediction, but current approaches have exclusively concentrated on addressing one of these issues, overlooking the similarity of both tasks in reflecting the relationship of protein sequences towards substrate structures.”

7. A discrepancy is observed between line 831 and Figure 5e regarding the representation of K_{cat}/K_m . The authors should rectify this inconsistency to ensure clarity.

Response:

We appreciate you for pointing out this error. In Figure 5, the correct notation should be k_{cat} / K_m . We have made the necessary corrections and thoroughly reviewed the content to address the inconsistency, as shown in Fig. R2.

Fig. R2 Scatter plot illustrating the Pearson coefficient correlation (PCC) between experimentally measured k_{cat} / K_m values and predicted k_{cat} / K_m values of UniKP for k_{cat} / K_m dataset (N=910). The color gradient represents the density of data points, ranging from blue (0.02) to red (0.28).

8. In Figure 5, a scatter plot depicting the predicted k_{cat} against the experimental K_m would provide a clearer visualization. Moreover, it is important to note that RMSE and R^2 (coefficient of determination) should not share the same Y-axis due to their different dimensions. RMSE measures the difference between model predictions and actual values and is expressed in the same units as the predicted and actual values. As UniKP predicts k_{cat} in s^{-1} and K_m is typically measured in mM, RMSE lacks practical significance. On the other hand, R^2 represents the proportion of variance in the dependent variable that can be predicted from the independent variable, and it is a dimensionless value between 0 and 1. Since R^2 describes linear correlation, it may be more suitable to include Spearman correlation for assessing the model's performance.

Response:

We sincerely apologize for the confusion in our previous presentation. The axes in Figure 5 should represent the experimental k_{cat} / K_m values on the x-axis and the predicted k_{cat} / K_m values on the y-axis. We have revised it as the Response 7.

Additionally, we have reorganized the graphical representation of RMSE and R^2 results for better understanding, as shown below.

Fig. R3 | RMSE, coefficient of determination (R^2) between experimentally measured K_m values and predicted K_m values on K_m test set.

We apologize for any misunderstanding on this section. The comparisons were only between the experimental and predicted k_{cat} values, or experimental and predicted K_m values, and not between k_{cat}

and K_m values. Here, we opted for RMSE and R^2 based on previous literature [3], where these metrics are commonly used for model performance evaluation in regression tasks.

9. In Figure 6a, AsTAL exhibits a higher k_{cat} than RgTAL, while another TAL enzyme, TALclu, demonstrates a k_{cat}/K_m four times higher than that of RgTAL (ChemBioChem 2022, 23, e2022000). This raises the question of why RgTAL was chosen, as it does not exhibit optimal enzymatic activity. It is important to clarify whether UniKP is limited to redesigning enzymes with mid-range k_{cat} values, while unable to further improve enzymes that already possess the highest catalytic activity.

Response:

Thanks for your careful consideration, when we select the TAL, we considered its industrial applications. Since its potential applications in biosynthesis occurred in neutral pH [4], we selected the most commonly used TAL with highest performance in metabolic engineering, which is RgTAL, whereas TALclu showed its optimal performance under alkaline condition (pH 9.5). Our results demonstrated the ability of UniKP to discover and evolve new enzymes that have better performance than RgTAL.

To further demonstrate the advantage of UniKP, especially EF-UniKP, we also conducted additional enzyme discovery experiments based on the TALclu.

Firstly, we examined whether we could identify better TAL mutant based on TALclu using UniKP. We performed a Blast search against the TALclu and then selected the top 1000 sequences with the highest similarity for prediction using UniKP without environmental factors. From the predictions, we selected the top 5 sequences with the highest predicted k_{cat} values for wet-lab experimental validation. Remarkably, we observed that one sequence achieved k_{cat} / K_m values 2.2 times higher than that of TALclu (Fig. R4).

Secondly, we examined whether we could identify better TAL mutant at pH 9.5 using EF-UniKP. We performed a Blast search against the TALclu and then selected the top 1000 sequences with the highest similarity for prediction using EF-UniKP, with a pH setting of 9.5. From the predictions, we selected the top 5 sequences with the highest predicted k_{cat} values for wet-lab experimental validation. Remarkably, we observed that the k_{cat} and k_{cat} / K_m values for all five selected sequences exceeded those of TALclu, with two of the sequences achieving k_{cat} / K_m values 2.5-2.6 times higher than that of TALclu (Fig. R5).

We have demonstrated that the UniKP could discover highly active Tyrosine ammonia lyase (TAL) enzymes compared to TALclu, as shown below. This further demonstrates the robust prediction capabilities of UniKP.

TALs	k_{cat} (s ⁻¹)	K_m (mM)	k_{cat} / K_m (s ⁻¹ ·mM ⁻¹)
TALclu	74.37	0.46	161.67
MgTAL	47.08	0.19	247.79
OsTAL	54.19	0.15	361.27
MpTAL	97.25	0.41	237.20
CsTAL	64.21	0.48	133.77
CnTAL	69.67	0.34	204.91

Fig. R4 The kinetic characteristics of wild-type Tyrosine ammonia lyase from *Chryseobacterium luteum* sp. nov (TALclu) and newly discovered TALs mined from non-redundant protein database by performing BLASTP. The top 5 sequences with the highest predicted k_{cat} values by UniKP were selected for experimental validation, including MgTAL from *Marinirhabdus gelatinilytica* (WP_115122402.1), OsTAL from *Oceanihabitans sediminis* (WP_113965696.1), MpTAL from

Myroides pelagicus (WP_155035460.1), CsTAL from *Chryseobacterium* sp. StRB126 (WP_045501302.1), CnTAL from *Chryseobacterium nakagawai* (WP_185145930.1). All the experiments were conducted under a pH of 8.5.

TALs	k_{cat} (s ⁻¹)	K_m (mM)	k_{cat} / K_m (s ⁻¹ ·mM ⁻¹)
TALclu	16.54	0.05	330.80
TrTAL	34.54	0.04	863.50
HiTAL	76.00	0.21	361.90
LeTAL	33.85	0.07	483.57
PpTAL	28.09	0.04	702.25
AaTAL	25.24	0.03	841.33

Fig. R5 The kinetic characteristics of wild-type Tyrosine ammonia lyase from *Chryseobacterium luteum* sp. nov (TALclu) and newly discovered TALs mined from non-redundant protein database by performing BLASTP. The top 5 sequences with the highest predicted k_{cat} values by UniKP were selected for experimental validation, including TrTAL from *Tephrocye rancida* (KAG6920185.1), HiTAL from *Heterobasidion irregulare* TC 32-1 (XP_009553370.1), LeTAL from *Lentinula edodes* (KAF8828722.1), PpTAL from *Pleurotus pulmonarius* (KAF4563271.1), AaTAL from *Aspergillus arachidicola* (KAE8337485.1). All the experiments were conducted under a pH of 9.5.

10. Lastly, the current description of Kcat determination lacks specificity. Including figures depicting the kinetic data of Kcat would enhance the credibility of the study.

Response:

We have added graphical representations of the experimental process and data related to k_{cat} determination for better understanding, as shown in Fig. R6.

Fig. R6 The kinetic curves of RgTAL and five mutants generated by UniKP. All variants were generated for all single-point RgTAL mutations, where each variant involves mutating an amino acid at a specific position to one of the 19 amino acids other than itself, which resulted in a total of variants equal to the product of 19 and the length of the sequence (19*693=13,167). Through an in-silico screening of all 13,167 single-point mutations in RgTAL using UniKP, ranked by k_{cat} / K_m values, the top 5 mutants were identified from each screening (k_{cat} / K_m) for experimental validation. And MT denotes the mutated form of RgTAL.

Taking this result as an example, we have included all kinetic curves of experimental results (Supplementary Fig. 13-16) in the Supplementary Information for reference.

I hope incorporating the suggested revisions and addressing the concerns and inquiries mentioned above will significantly improve the manuscript.

References:

- [1] Yu, Han, et al. "IPPF-FE: an integrated peptide and protein function prediction framework based on fused features and ensemble models." *Briefings in Bioinformatics* 24.1 (2023): bbac476.
- [2] Kroll, Alexander, and Martin J. Lercher. "Machine learning models for the prediction of enzyme properties should be tested on proteins not used for model training." *bioRxiv* (2023): 2023-02.
- [3] Li, Feiran, et al. "Deep learning-based k cat prediction enables improved enzyme-constrained model reconstruction." *Nature Catalysis* 5.8 (2022): 662-672.
- [4] Zhou, Shenghu, Tingting Hao, and Jingwen Zhou. "Fermentation and metabolic pathway optimization to de novo synthesize (2S)-naringenin in *Escherichia coli*." *Journal of Microbiology and Biotechnology* 30.10 (2020): 1574.

Reviewer #3 (Remarks to the Author):

In the manuscript presented by Yu, et al., a computational framework for prediction of kinetic parameters of enzymes, namely, turnover numbers (k_{cat}), Michaelis constants (K_m) and enzyme efficiency (k_{cat}/K_m), UniKP, is presented. This framework is based on pre-trained language models that use protein sequence and substrate structures as input features. Additionally, a two-layer machine learning approach is used for accounting for pH and temperature variations in predicted parameters, and different approaches for improving prediction of high k_{cat} values are tested. The authors ran extensive comparison on predictive performance of their model vs. the state-of-the-art tools for prediction of k_{cat} (Li et al. 2022) and K_m (Kroll et al. 2023) parameters, showing improvements in correlation coefficients and error metrics. However, the prediction errors displayed by UniKP are still above an order of magnitude in kinetic parameters, on average, which limits large-scale applicability for accurate modeling of cellular function, leaving room for further improvements.

Notably, the authors show how this tool can be used for mining sequences and selecting the optimal enzyme for a specific metabolic reaction (tyrosine ammonia liase, TAL), and predict single-point mutations that increase enzymatic performance significantly, achieving the highest reported enzyme efficiency for a TAL mutant.

The subject addressed by this study is of current high relevance in the fields of synthetic and systems biology, machine learning and metabolic engineering, therefore, the results presented here are of high relevance for a wide public. The use of pre-trained language models for prediction of enzyme parameters offers a new perspective on the use of machine learning methods for biological purposes, the use of this method for enzyme discovery and directed improvement is a major novel contribution to the aforementioned fields, opening possibilities for many applications.

I recommend the authors to address the following major and minor points in order to improve the quality of this manuscript and support their findings even further.

Response:

We are immensely grateful for the thorough evaluation of our manuscript and the constructive comments about our work which helped us improve the quality of our manuscript. We have carefully revised the manuscript according to the comments.

Major points:

1.- Most of the model performance evaluations in this manuscript focus on the comparison of correlation, determination coefficients and error metrics with the previously published DLKcat. Recently, a preprint highlighting the flaws in the selection and partitioning of the DLKcat dataset (<https://doi.org/10.1101/2023.02.06.526991>) has been published in bioRxiv. This study demonstrates that DLKcat predictions are accurate just for enzymes that are highly similar to those used in the training dataset. They show that a simple approach of approximating K_{cats} , by computing geometric means of k_{cats} across the 3 top-similar enzymes (sequence similarity) in the training dataset is capable of yielding significantly better determination coefficients for predictions of the test dataset. I recommend the authors to also run a comparison of this type, as the predictions of DLKcat have already been found to be biased.

Response:

Thank you for your valuable suggestion. We appreciate your input and have taken your advice into consideration. We have conducted additional experiments as you suggested. Specifically, we made a comparison based on different sequence similarity, comparing it with predictions generated using the geometric means (gmean) of k_{cat} values from the three most similar enzymes, as outlined in the reference (<https://doi.org/10.1101/2023.02.06.526991>). The results of our comparative analysis showed a significant advantage compared with gmean on different intervals, as shown in Fig. R1.

Fig. R1 Performance Comparison of UniKP and geometric mean (gmean). a, b) The curves are coefficients of variation R^2 , RMSE, calculated in sliding windows of size $n=100$ across sequences in the test set ordered by the maximal sequence identity between individual test enzymes and all sequences in the training data. Position on the x-axis indicates the mean across the window. Red: UniKP predictions; yellow: geometric mean of k_{cat} values, calculated over the three most similar enzymes in the training set. The two points at the top right are for test datapoints with enzymes already used for training (100% max. sequence identity), these were not included in the sliding windows.

We have also added the following sentences in the Results section of the revised manuscript:

“In order to evaluate the potential data leakage, we used a reported method to compare the performance of UniKP with the geometric mean of experimental data which represent the potential data leakage [30]. The result demonstrated a clear advantage (higher correlation and lower RMSE) of UniKP over the geometric means, indicating the absence of data leakage in the training process (Supplementary Fig. 3).”

2.- Additionally, as major part of the results and discussion focus on comparison with DLKcat, the large error metrics displayed by uniKP. are not discussed in detail This is common in AI-ML literature, where performance is usually evaluated in terms of correlation or model fitness. In this case, the AI framework

proved to be useful for directed enzyme improvement. Nonetheless, readily available kinetic parameters in a large-scale has been one of the most important bottlenecks for advancing on mechanistic understanding of cell behavior through modeling attempts. For this whole field of biological sciences, it is crucial to provide accurate parameters that enable appropriate quantification of cellular states and dynamics. AI-ML methods offer a great opportunity for addressing this long-standing issue, but more development is still needed. This manuscript would benefit from a broader and deeper discussion on the limitations of accurate quantitative prediction of kinetic parameters and their implications for bottom-up biological studies.

Response:

We greatly appreciate your insightful suggestions. We have taken your advice and expanded our discussion to encompass the limitations of quantitative prediction of kinetic parameters and the broader implications for bottom-up biological studies in our manuscript. We have added the following sentences in the last paragraph:

“A central objective in synthetic biology is the development of a digital cell, poised to revolutionize our methods of studying biology. A critical prerequisite for this endeavour is the meticulous determination of enzymatic parameters for all enzymes within the pathway. Tools assisted by artificial intelligence illuminate this challenge, offering a high-throughput approach to predicting enzymatic kinetics. However, despite the reduced errors in UniKP predictors compared to earlier models, inaccuracies remain a significant hurdle in crafting a precise metabolic model. The inclusion of a growing number of experimentally determined k_{cat} and K_m values, sourced from cutting-edge high-throughput experimental techniques like those employed in modern biofoundries, can enhance model accuracy.”

3.- The use of a machine learning model for Kcat prediction is justified in the first results subsection by showing that the concatenated vectors (combined representation of sequence and substrate) cannot differentiate between Kcat values of different orders of magnitude, by projecting such vectors using t-SNE analysis. In the methods section it is mentioned that such analysis was ran using just the default parameters. It is not mentioned which was the specific software implementation used for this, and default parameters may differ from one to another. Moreover, t-SNE is a stochastic method that relies on two main parameters (perplexity and number of iterations). The choice of these parameters is crucial, as they may have drastic effects on the projection results. In order to strengthen this section, I recommend the authors to run a parametric analysis of the t-SNE projections to show the effect of the parameters in the grouping of the concatenated enzyme vectors. If Kcat values do not seem to group despite the chosen perplexity and number of iterations, then the implementation of uniKP will be better justified.

Response:

Thank you for your valuable suggestion. We have tested different perplexity and iterations and added them to the Supplementary information, as shown in Fig. R2. We have also revised the main text as followed:

“The concatenated representation vector of both the protein and substrate are then fed into the following machine learning module. Here, the projections of t-distributed Stochastic Neighbour Embedding (t-SNE) with different perplexity and iterations demonstrated that a solely concatenated representation vector cannot discriminate well between high and low k_{cat} values [26], further emphasizing the need for the machine learning model (Supplementary Fig. 1).”

Fig. R2. Scatter plots showing t-distributed Stochastic Neighbor Embedding (t-SNE) for different perplexity values (10, 20, 30, 40, 50) and iterations (250, 500, 750, 1000) using the DLKcat dataset. The color gradient represents experimentally measured kcat values (logarithm with base 10) of data points, ranging from blue (-7) to red (7). The embedded vectors have been normalized to a range of 0 to 1.

Minor points:

4.- Lines 47-48: The applications and implications of this study focus a lot on the use of this method for enhancing enzyme engineering, however, large-scale and accurate prediction of kinetic parameters could also majorly benefit systems biology, providing better parameters for quantitative modeling studies aiming to understand biological networks, specially metabolism. The manuscript could benefit from adding this point here.

Response:

We agree with the reviewer's opinion about the applications of this method in systems biology and its associated benefit. Although enzyme engineering is one important topic in synthetic biology, the

quantitative modelling could help us to better understand the metabolic networks. However, after our discussion in person with authors of the DLKcat paper, who had applied the predicted parameters to construct models, we believe it is still premature for constructing an accurate model at the current status. As the response in point 2, despite the reduced errors in UniKP predictors compared to earlier models, inaccuracies remain a significant hurdle in crafting a precise metabolic model. We have added the relevant discussion in the last paragraph as in point 2.

5.- Lines 52-54: It is mentioned that, in comparison to the availability of millions of sequences, uniprot just offers around 2,000 kcat values. On the other hand, tenths of thousands of kinetic parameters are available in BRENDA and SABIO-RK bases, and in their latest versions these databases have integrated uniprot identifiers to their entries, facilitating a larger connection between measured parameters and protein sequences. Please mention this, or even add the corresponding numbers if possible, in order to clarify this better for the readers.

Response: We apologize for the unclear description. We have taken your feedback into consideration and have made revisions to the manuscript by including additional information about these database in our manuscript, as shown below.

“For instance, the sequence database UniProt contains over 230 million enzyme sequences, while enzyme databases BRENDA and SABIO-RK contain tens of thousands of experimentally measured k_{cat} values [6-8]. The integration of UniProt identifiers in these enzyme databases facilitated the connection between measured parameters and protein sequences. However, these connections are still far smaller in scale compared to the number of enzyme sequences, limiting the advancement of downstream applications such as directed evolution and metabolic engineering.”

6.- Lines 69-70: “...often deviate from the ground truth”. What is such truth regarding enzyme parameters, in vitro measurements? in vivo values? (probably impossible to characterize), predictions from basic principles using quantum chemistry? Even more, modern models of science and its methods of discovery do not aim to unveil “truths” from nature, but to create falsifiable hypotheses that can be empirically tested, and proof to offer coherent and viable explanations that expand our understanding of nature and our ways to modify it/interact with it. Please avoid the use of these categorical terms.

Response:

We greatly appreciate your insightful and thoughtful comments. We have revised it, as shown below.

“Consequently, the calculated k_{cat} / K_m values from these models often deviate significantly from the experimental measurements.”

7.- Overall in the introduction and abstract sections prediction of kinetic parameters is mentioned repeatedly, but it is never said that this refers to prediction of in vitro measured parameters, please clarify this where at least once in these sections.

Response:

We apologize for this unclear description, we have added this statement in the introduction of our revised manuscript, as shown below.

“The *in vitro* measured values of k_{cat} and K_m , the maximal turnover rate and Michaelis constant, are the indicators of the efficiency of an enzyme in catalyzing a specific reaction and can be used to compare the relative catalytic activity of different enzymes [4].”

8.- Line 140: “further demonstrating the superiority of UniKP”. The top used synonyms of the word superiority are dominance, excellence and perfection, which I doubt that is what the authors wanted

to express. UniKP shows improved performance in comparison to other frameworks, major in some aspects, but as any approximation it is also limited in its prediction power, I recommend to avoid the use of this charged term here and in other parts of the text. Science is not a competition, but rather a cumulative enterprise of humanity for understanding nature.

Response:

We have greatly benefited from your insightful comments. We have checked and revised our manuscript, as shown below.

“Additionally, the highest value of DLKcat in these five rounds was 16% lower than the lowest value of UniKP, further demonstrating the robustness of UniKP.”

9.- Line 154: “Theoretically, the former should exhibit higher values”. This has been explored by many studies before. Please add references.

Response: Thanks for your careful check. We have added it in our manuscript, as follows.

“Theoretically, the former should exhibit higher values [27-28].”

10.- Line 154: “Our results revealed”, as mentioned above, many studies have explored this issue before, using different approaches, therefore, this manuscript is not revealing this but confirming or showing that results agree with what has been previously reported. Please modify accordingly.

Response:

We greatly appreciate your meticulous review, and we have revised it, as shown below.

“Our results found that the primary central and energy metabolism category was significantly higher than the latter, consistent with expectations (Fig. 2e; $p = 9.33 \times 10^{-8}$)”

11.- Lines 171-176: Could the authors add any error metrics here? (RMSE, or median absolute errors). Enzyme parameters are crucial for quantitative studies, hence, provide a measure of the associated errors in predictions will inform the community better.

Response:

Thanks for your careful consideration, we have added it and checked our manuscript to ensure that error metrics are all included, as shown below.

“Moreover, we obtained R^2 values of 0.60 for wild-type enzymes and 0.81 for mutant enzymes, along with RMSE values of 0.90 for wild-type enzymes and 0.67 for mutant enzymes.”

12.- Line 182-183: Could the authors elaborate on why the selected enzyme-substrate pair is a crucial one and what is its relevance for the field?

Response:

We intended to provide an illustrative example to verify the advantages of UniKP. However, we agree with your point that it is not necessary to include a single case lacking significant relevance to this field. Therefore, we have removed it in our manuscript.

13.- Line 236: “predicting high kcat values was essential”, these predictions are still gonna be essential for the field, even after publication of this manuscript. These predictions are very challenging and more accurate frameworks for prediction are likely to appear in the upcoming years, with further understanding and expansion oof machine learning methods. Please change the word “was” to has been or something similar.

Response:

We hold great admiration for your rigorous check, and we have made the corresponding modifications as your suggestions, as shown below.

“However, predicting high k_{cat} values has been essential in enzymology and synthetic biology [14-15]”

14.- Line 316: “prediction framework, UniKP, which can accurately predict three essential enzyme kinetic parameters”. UniKP offers improvements in comparison with previous methods. Nevertheless, its predictions may differ from in vitro measurements even by an order of magnitude, which questions the use of the term “accurately predict” in absolute terms. It would be more fair to mention that it improves accuracy of predictions.

Response:

Thank you for your valuable comment. We have checked and revised this statement in our manuscript, as shown below.

“Here, we present a pretrained language model-based enzyme kinetic parameters prediction framework (UniKP), which improves accuracy of predictions for three enzyme kinetic parameters, k_{cat} , K_{m} , and $k_{\text{cat}} / K_{\text{m}}$, from a given enzyme sequence and substrate structure.”

15.- Line 337: “As the learnable dataset expands”. Here the authors acknowledge that the available measurements will expand, which I agree. It is even likely that future techniques will enable high-throughput characterization of enzymes. Nonetheless, in most of the text, the tone gives the idea that high-throughput prediction is the only active direction in this topic. Adding some nuances here and probably citing latest research on experimental methods and research directions would benefit the discussion.

Response:

Thank you for your valuable comment. We have taken your suggestion into account and provided a more balanced perspective by incorporating some recent research on experimental methods and ongoing research directions to enhance the depth of the discussion, as shown below.

“As experimental techniques advance, including biofoundry lab automation and continuous evolution methods [41-42], we anticipate a surge in enzyme kinetic data. This influx will not only enrich the field but also enhance the accuracy of prediction models.”

Reviewer #1 (Remarks to the Author):

All my queries have been well addressed.

Reviewer #2 (Remarks to the Author):

I have reviewed the revised manuscript and find that the authors have addressed many of the concerns raised by the reviewers, including mine. It is clear that the authors have undertaken significant efforts to improve the quality of their manuscript in response to the initial round of reviews. However, there are still some issues that need to be resolved. My concerns are detailed below:

1. Data Leakage and Comparison Methodology:

A lingering concern pertains to the issue of data leakage and the comparison with other methods to elucidate the contributions of this work. It is praiseworthy that the authors aligned their methodology with a preprint and a paper published in Nature Communications (Nat Commun 14, 4139, 2023) to facilitate direct comparisons. During the revision, the authors employed the geometric mean of the closest 100 sequences in the dataset, citing Kroll et al. However, it should be noted that Kroll et al. used the 3 most similar sequences for their calculations. I would recommend that the authors follow the same setting for a more accurate and fair comparison. Also, the values for UniKP and gmean in the 50–70% mean sequence identity region are conspicuously alike. A discussion or clarification of this would add value to the manuscript. Lastly, if possible, a head-to-head comparison with TurNuP would enhance the manuscript's validation further.

2. Correlation between Predictors:

Between lines 308 and 310, the authors state that the Pearson Correlation Coefficient (PCC) between prediction and measurement is -0.02 when using other state-of-the-art methods. This point would benefit from further elaboration. Specifically, can the authors present the results of the correlation between UniKP Kcat/ UniKP Km and the Kcat/Km dataset? Additionally, a more thorough discussion about the current failure of using Kcat and Km predictors for the Kcat/Km values would be valuable.

Reviewer #3 (Remarks to the Author):

The original reviewer was unable to comment. The comments were provided by one of the other reviewers.

For point #2, the limitations of accurate quantitative prediction and the significance of the accurate model for bottom-up biological studies have been discussed in the revised manuscript as the reviewer suggested.

For point #3, the results provided by the authors show that Kcat values do not group in t-SNE analysis with different parameters, which justify the implementation of uniKP.

In general, the concerns raised by the reviewer have been addressed in my opinion.

Response to Reviewer's Comments:

We express our sincere gratitude to all reviewers for their thorough and insightful feedback, and for giving us an opportunity to further revise our manuscript. In light of their comments, we have carefully revised the manuscript and included additional data where necessary. Below, we address each of the comments in detail.

Reviewer comments

Reviewer #1 (Remarks to the Author):

All my queries have been well addressed.

Response:

We appreciate the reviewer's thoughtful evaluation of our manuscript.

Reviewer #2 (Remarks to the Author):

I have reviewed the revised manuscript and find that the authors have addressed many of the concerns raised by the reviewers, including mine. It is clear that the authors have undertaken significant efforts to improve the quality of their manuscript in response to the initial round of reviews. However, there are still some issues that need to be resolved. My concerns are detailed below:

Response:

We appreciate the reviewer's thoughtful evaluation of our manuscript and the constructive comments about our work which further helped us improve the quality of our manuscript. We have carefully revised the manuscript according to the comments.

1. Data Leakage and Comparison Methodology:

A lingering concern pertains to the issue of data leakage and the comparison with other methods to elucidate the contributions of this work. It is praiseworthy that the authors aligned their methodology with a preprint and a paper published in Nature Communications (Nat Commun 14, 4139, 2023) to facilitate direct comparisons. During the revision, the authors employed the geometric mean of the closest 100 sequences in the dataset, citing Kroll et al. However, it should be noted that Kroll et al. used the 3 most similar sequences for their calculations. I would recommend that the authors follow the same setting for a more accurate and fair comparison.

Response:

We apologize for any confusion caused by unclear description. To clarify, our approach is exactly the same as the methods detailed in the bioRxiv manuscript, where the R^2 of geometric mean of the k_{cat} values of the 3 most similar enzymes for each enzyme in the training dataset (represent the potential leaking information) was compared with corresponding UniKP predicted values. And the curves are R^2 , calculated in sliding windows of size $n=100$ across sequences in the test set ordered by the maximal sequence identity between individual test enzymes and all sequences in the training data, as previously outlined in the same bioRxiv manuscript.

We have also provided explanations in the legend of Supplementary Fig. 3.

“Supplementary Figure 3. a, b) Performance Comparison of UniKP and geometric mean (gmean). The curves are coefficients of variation R^2 (a), RMSE (b), calculated in sliding windows of size $n=100$ across sequences in the test set ordered by the maximal sequence identity between individual test enzymes and all sequences in the training data. Position on the x-axis indicates the mean across the window. Red: UniKP predictions; yellow: geometric mean of k_{cat} values, calculated over the three most similar enzymes in the training set. The two points at the top right are for test datapoints with enzymes already used for training (100% max. sequence identity), these were not included in the sliding windows.”

Also, the values for UniKP and gmean in the 50–70% mean sequence identity region are conspicuously alike. A discussion or clarification of this would add value to the manuscript.

We appreciate the reviewer's astute observation regarding the similarity in values between UniKP and gmean within the 50–70% mean sequence identity region. Upon further examination, we undertook a meticulous analysis to elucidate this phenomenon, which encompassed an investigation into data distribution, potential data leakage, and the limitations inherent in the method employed. Our conclusion posits that within the realm of biochemistry and enzymology, particularly in the pursuit of alternative enzyme discovery, researchers frequently use BLAST analysis to identify closely related sequences (with a sequence identity range of 50-95%) under the assumption of potential functional similarity albeit with nuanced differences in activities or substrate specificities. It is plausible that the three most analogous sequences harbor a certain predictive capacity for kinetic parameters, hence the observed similarity between the gmean curve and the UniKP values within the specified identity range. This commonality in machine learning tasks is expected as the predictive values invariably reflect the characteristics of the most similar samples. However, the discernible difference between the two curves accentuates the enhanced depth of information captured by UniKP, displaying a notable advantage over the simple gmean prediction.

The divergences in curves at low identity regions (<50) and high identity regions (>70) warrant further discussion. In the low identity region, due to the scant sequence similarity, there is a higher probability of the most similar sequences pertaining to entirely distinct enzyme groups with divergent functions, thereby resulting in significantly lower correlations in gmean. Conversely, in the high identity region, the training dataset encompasses many enzyme mutants with a broad spectrum of kinetics, often generated from mutational experiments, which consequently lowers the gmean correlations. The higher correlation of UniKP predicted values in these regions underscores the capability of UniKP in discerning deeper interconnected information, thus exhibiting enhanced performance amidst seemingly contradictory data. The successful application of UniKP in identifying better enzyme single point mutants further substantiates our conclusion, as potential data leakage would invariably impede the accuracy of such predictive tasks.

We have incorporated the following text in the discussion section to reflect this analysis:

“The disparity between the high identity region (>70 identity) and low identity region (<50 identity) of the R^2 of UniKP predicted values and R^2 of the gmean method underscores the adeptness of UniKP in extracting deeper interconnected information, thereby demonstrating a higher predictive accuracy in these tasks.”

Lastly, if possible, a head-to-head comparison with TurNuP would enhance the manuscript's validation further.

We agreed with the reviewers that more comparison with related methods would be better to validate our method, however, TurNuP used different dataset with additional reaction information, which hindered the head-to-head comparison. In the future, we also plan to add reaction information in UniKP to enable it with improved accuracy and additional functions.

2. Correlation between Predictors:

Between lines 308 and 310, the authors state that the Pearson Correlation Coefficient (PCC) between prediction and measurement is -0.02 when using other state-of-the-art methods. This point would benefit from further elaboration. Specifically, can the authors present the results of the correlation between UniKP Kcat/ UniKP Km and the Kcat/Km dataset? Additionally, a more thorough discussion about the current failure of using Kcat and Km predictors for the Kcat/Km values would be valuable.

Response:

Thanks for your valuable comment. As the response to the first issue from reviewer 1, we conducted an additional detailed analysis and discussion.

We validated the UniKP framework based on experimentally measured k_{cat} / K_m values and k_{cat} / K_m values calculated using k_{cat} and K_m prediction models on k_{cat} / K_m dataset. In the case of k_{cat} and K_m prediction models, their corresponding training data came from different sources with different curation standards, which may result in inconsistency, posing challenges in achieving great performance when the predicted values were used for further calculation (PCC = -0.01 between experimentally measured k_{cat} / K_m values and k_{cat} / K_m values calculated using k_{cat} and K_m prediction models on k_{cat} / K_m dataset).

However, if the same set of data was used to train both models, the calculation should work. To demonstrate this, we retrieved 658 samples containing k_{cat} / K_m values from the BRENDA database, all of which were presented in the k_{cat} dataset. The corresponding K_m value can be calculated. Subsequently, we performed 5-fold cross-validation to independently predict the values of k_{cat} and K_m for all samples, and then calculated the corresponding k_{cat} / K_m values for each sample. We observed a higher correlation coefficient of 0.64 between the calculated k_{cat} / K_m values and experimentally measured k_{cat} / K_m . This further confirms that our proposed UniKP framework can indeed ensure consistent predictions when the data is consistent. Although the calculation has some accuracy, the PCC is still lower than prediction with k_{cat} / K_m dataset owing to its relatively limited sample size. Therefore, we have presented different prediction models for each enzyme kinetic parameters to provide more insights for various downstream tasks.

We have added the following texts in the discussion section:

“Furthermore, we validated the UniKP framework based on experimentally measured k_{cat} / K_m values and k_{cat} / K_m values calculated using k_{cat} and K_m prediction models on k_{cat} / K_m dataset. It is to be noted that the correlation observed between the values derived from UniKP k_{cat} / K_m and the experimental k_{cat} / K_m is relatively low (PCC=-0.01). This discrepancy is likely attributable to the disparate datasets employed in constructing the respective models, necessitating the development of a distinct model for predicting k_{cat} / K_m values. In the future, with the availability of a unified dataset encompassing both k_{cat} and K_m values, it is anticipated that the calculated outputs from the k_{cat} and K_m models would closely align with those generated by a dedicated model for k_{cat} / K_m .”

Reviewer #3 (Remarks to the Author):

The original reviewer was unable to comment. The comments were provided by one of the other reviewers.

For point #2, the limitations of accurate quantitative prediction and the significance of the accurate model for bottom-up biological studies have been discussed in the revised manuscript as the reviewer suggested.

For point #3, the results provided by the authors show that K_{cat} values do not group in t-SNE analysis with different parameters, which justify the implementation of uniKP.

In general, the concerns raised by the reviewer have been addressed in my opinion.

Response:

We appreciate the reviewer's thoughtful evaluation of our manuscript.

Reviewer #2 (Remarks to the Author):

All my concerns have been addressed

Response to Reviewer's Comments:

We express our sincere gratitude to all reviewers for their thorough and insightful feedback. Below, we address each of the comments in detail.

Reviewer comments

Reviewer #2 (Remarks to the Author):

All my concerns have been addressed.

Response:

We appreciate the reviewer's thoughtful evaluation of our manuscript.